

# Model simulations of the chemical and aerosol microphysical evolution of the Sarychev Peak 2009 eruption cloud compared to in-situ and satellite observations

Thibaut Lurton[1,6], Fabrice Jégou[1], Gwenaël Berthet[1], Jean-Baptiste Renard[1], Lieven Clarisse[2], Anja Schmidt[3,4], Colette Brogniez[5], and Tjarda Roberts[1]

[1]LPC2E/CNRS/Université d'Orléans, 3A, avenue de la Recherche Scientifique, F-45071 Orléans Cedex 2, France
[2]CQP/Université Libre de Bruxelles, CP160/09, avenue F. D. Roosevelt 50, B-1050 Brussels, Belgium
[3]Department of Chemistry, University of Cambridge, Lensfield Road, Cambridge CB2 1EW, United Kingdom
[4]Department of Geography, University of Cambridge, Downing Place, Cambridge CB2 3EN, United Kingdom
[5]Laboratoire d'Optique Atmosphérique, Université Lille 1, Cité Scientifique, F-59655 Villeneuve d'Ascq Cedex, France
[6]Now with IPSL/CNRS, 4, place Jussieu, F-75252 Paris Cedex 05, France

*Correspondence to:* T. Lurton (thibaut.lurton@ipsl.fr)

**Abstract.** Volcanic eruptions impact climate through the injection of sulfur dioxide ($SO_2$), which is oxidized to form sulfuric acid aerosol particles that can enhance the stratospheric aerosol optical depth (SAOD). Besides large-magnitude eruptions, moderate-magnitude eruptions such as Kasatochi in 2008 and Sarychev Peak in 2009 can have a significant impact on stratospheric aerosol and hence climate. However, uncertainties remain in quantifying the atmospheric and climatic impacts of the

2009 Sarychev Peak eruption due to limitations in previous model representations of volcanic aerosol microphysics and particle size, whilst biases have been identified in satellite estimates of post-eruption SAOD. In addition, the 2009 Sarychev Peak eruption co-injected hydrogen chloride (HCl) alongside $SO_2$, whose potential stratospheric chemistry impacts have not been investigated to date. We present a study of the stratospheric $SO_2$-particle-HCl processing and impacts following Sarychev Peak eruption, using the CESM1(WACCM)-CARMA sectional aerosol microphysics model (with no a priori assumption on

particle size). The Sarychev Peak 2009 eruption injected $0.9\,\text{Tg}$ of $SO_2$ into the upper troposphere and lower stratosphere (UTLS), enhancing the aerosol load in the Northern hemisphere. The post-eruption evolution of the volcanic $SO_2$ in space and time are well reproduced by the model when compared to IASI (Infrared Atmospheric Sounding Interferometer) satellite data. Co-injection of 27 Gg HCl causes a lengthening of the $SO_2$ lifetime and a slight delay in the formation of aerosols, and acts to enhance the destruction of stratospheric ozone and mono-nitrogen oxides ($NO_x$) compared to the simulation with volcanic $SO_2$

only. We therefore highlight the need to account for volcanic halogen chemistry when simulating the impact of eruptions such as Sarychev on stratospheric chemistry. The model-simulated evolution of effective radius ($r_{\text{eff}}$), reflects new particle formation followed by particle growth that enhances $r_{\text{eff}}$ to reach up to $0.2\,\mu\text{m}$ on zonal average. Comparisons of the model-simulated particle number and size-distributions to balloon-borne in-situ stratospheric observations over Kiruna, Sweden, in August and September 2009, and over Laramie, U.S.A., in June and November 2009 show good agreement and quantitatively confirms

the post-eruption particle enhancement. We show that the model-simulated SAOD is consistent with that derived from OSIRIS (Optical Spectrograph and InfraRed Imager System) when both the saturation bias of OSIRIS and the fact that extinction pro-



files may terminate well above the tropopause are taken into account. Previous modelling studies (involving assumptions on particle size) that reported agreement to (biased) post-eruption estimates of SAOD derived from OSIRIS likely underestimated the climate impact of the 2009 Sarychev Peak eruption.

## 1 Introduction

Explosive volcanic eruptions inject large quantities of sulfur dioxide ($SO_2$) into the atmosphere and have the potential to affect global climate (McCormick et al., 1995; Robock, 2000). Volcanic eruptions impact the global radiative budget via the formation of sulfuric acid aerosol particles from the volcanic $SO_2$ emitted. The presence of this particle load at stratospheric altitudes enhances the stratospheric aerosol optical depth (SAOD) and increases the solar backscatter, thereby inducing a cooling at the Earth's surface. The lifetime of sulfuric acid aerosol particles in the stratosphere can reach several years, significantly longer

than in the troposphere (days to weeks). Large-magnitude eruptions that inject $SO_2$ directly into the stratosphere therefore typically have more prolonged and widespread (global or hemispheric) impacts than small-magnitude eruptions that typically inject $SO_2$ into the troposphere only. The June 1991 eruption of Mount Pinatubo was a large-magnitude eruption, with a Volcanic Explosivity Index (VEI, as defined in Newhall and Self (1982)) of 6, that had a significant impact on the stratospheric aerosol layer and hence climate (Bluth et al., 1992; Sato et al., 1993; Ammann et al., 2003): global visible AOD was enhanced

by up to 0.15, causing a surface cooling of up to 0.5 degrees Celsius (Douglass and Knox, 2005; Wunderlich and Mitchell, 2017). In addition, stratospheric halogens (bromine and chlorine, that are present at elevated post-industrial concentrations in the stratosphere as consequence of past anthropogenic chlorofluorocarbon (CFC) emissions) became activated through reactions on the volcanic aerosol, causing substantial depletion of stratospheric ozone and a larger Antarctic ozone hole (Portmann et al., 1996; Solomon et al., 1996).

Moderate-magnitude explosive volcanic eruptions may also reach the stratosphere, especially at high latitudes, where the tropopause height is lower. However, they typically have a much-reduced effect on climate and atmospheric chemistry compared to large-magnitude eruptions (Oman et al., 2005; Kravitz et al., 2010). This is for several reasons. First, in general a smaller mass of $SO_2$ is injected and oxidized to sulfate aerosol. Second, by injecting to lower altitudes, the emissions from moderate-magnitude eruptions are more susceptible to removal by stratospheric-tropospheric exchange processes. Third, the

impacts from moderate-magnitude eruptions at high-latitudes tend to be limited to one hemisphere only, in contrast to large eruptions at the tropics that inject $SO_2$ and aerosol into both hemispheres. Nevertheless, as volcanic eruption frequency follows an inverse power law with magnitude (e.g. Sparks (2003) and references therein), the cumulative impacts of frequent moderate-magnitude eruptions on stratospheric aerosol can be significant (Vernier et al., 2011), and for example were identified as a factor in recent decadal climate trends (Solomon et al., 2011; Ridley et al., 2014).

Here we study the moderate-magnitude eruption of Sarychev Peak volcano, which erupted in mid-June 2009 in the Kuril Islands, Russia (48°N; 153°E; 1512 m a.s.l.), injecting $SO_2$ and also HCl to the stratosphere. The eruption was classified with a VEI of 4 (the volcanic eruptive index as defined in Newhall and Self (1982) is a logarithmic scale based on the volume of tephra ejected), and the main volcanic emission was injected to heights around 9–14 km as estimated from IASI retrievals



(Carn et al., 2016; Carboni et al., 2016). Remote sensing observations over Eureka, Canadian Arctic, showed volcanic aerosol layers from the tropopause up to 16–17 km one month after the eruption, that subsequently settled into a more homogeneous layer in the lower stratosphere (O'Neill et al., 2012).

Previous modelling studies of the Sarychev 2009 eruption focused on the injection of $SO_2$, formation of volcanic aerosol and its radiative and atmospheric chemistry impacts (Haywood et al., 2010; Kravitz et al., 2011; Berthet et al., 2017). However, the models did not explicitly simulate the aerosol microphysical evolution of the volcanic cloud, rather they used bulk aerosol schemes and/or assumed size distributions. Model-simulated atmospheric impacts of the eruption on, for instance, aerosol optical depth, are highly dependent on the prescribed aerosol size (or effective radius, $r_{eff}$). An $r_{eff}$ of $0.13\,\mu m$ was assumed in the HadGEM2 model study by Haywood et al. (2010), whilst Kravitz et al. (2011) used scaling to adjust their ModelE (Schmidt et al., 2006) simulations to represent a similar $r_{eff}$. Measurements that locally quantified the post-eruption volcanic aerosol include ground-based remote sensing (Haywood et al., 2010; Kravitz et al., 2011; Mattis et al., 2010; O'Neill et al., 2012) and balloon-borne observations. For example Kravitz et al. (2010) and Jégou et al. (2013) present in-situ balloon-borne observations of size-resolved stratospheric aerosol over Laramie, Wyoming, U.S.A. (June and November, 2009) and Kiruna, Sweden (August-September, 2009), respectively. The estimates of $r_{eff}$ from all these measurements range from $0.1\,\mu m$ to $0.3\,\mu m$, i.e. larger than assumed in the models.

O'Neill et al. (2012) highlight that this discrepancy can translate into large uncertainties in the modelled impacts, e.g. doubling of particle size from that assumed by Haywood et al. (2010) would lead to five-fold increase in the hemispherical (per particle) backscattering cross section of sulfate particles.

Volcanic aerosol from the Sarychev eruption also affected stratospheric halogen chemistry, via heterogeneous reactions on the aerosol surface area. The impacts were more modest than found for large-magnitude eruptions such as 1991 Mt. Pinatubo, but simulations suggest ozone depletion up to 4% in the lower stratosphere at high latitudes, with local $NO_2$ depletion up to 40% (Berthet et al., 2017), consistent with balloon-based and satellite observations (Adams et al., 2017).

To evaluate and tune the models, studies to date have relied upon satellite data from the OSIRIS instrument (Optical Spectrograph and Infrared Imaging System) to provide a global estimation of aerosol optical depth. However, comparison between OSIRIS and the models found a $\approx 1$ month discrepancy in the timing of the SAOD maximum following the eruption. This was attributed to be likely due to deficiencies in the model aerosol microphysics, specifically the absence of nucleation processes (Haywood et al., 2010; Jégou et al., 2013). In subsequent work, Fromm et al. (2014) identified that stratospheric AOD derived from OSIRIS under high aerosol loadings was likely underestimated following volcanic eruptions, due to a saturation effect and because the extinction profiles may terminate well above the tropopause (and therefore miss volcanic aerosol in the lowermost stratosphere). As model studies to date have used OSIRIS-derived AOD's to evaluate and justify choice of model aerosol parameters such as $r_{eff}$ (Haywood et al., 2010; Kravitz et al., 2011) this finding invokes the need to re-examine the assumed volcanic aerosol properties in the models. Finally, there have also been recent advances in satellite observations of volcanic gases in the stratosphere. First, new retrievals now enable an improved estimation of $SO_2$ mass injected combined with estimates of plume height from IASI (Infrared Atmospheric Sounding Interferometer) on the MetOp-A satellite (Clarisse et al., 2012; Carboni et al., 2016). Second, recent analysis of satellite data from the Microwave Limb Sounder (MLS) onboard



satellite AURA identifies that Sarychev volcano co-injected HCl alongside $SO_2$ to the stratosphere (Carn et al., 2016). The co-injection of volcanic halogens alongside $SO_2$ could modify the resulting atmospheric chemistry/aerosol processing and impacts. In light of these advances it is instructive to perform a new model-observation study of the Sarychev 2009 eruption and its stratospheric impacts that furthermore benefits from recently developed model capabilities to simulate aerosol microphysics and size evolution.

Here we present model simulations of stratospheric aerosol evolution and chemistry following the moderate-magnitude 2009 Sarychev eruption using the global Community Earth System Model (CESM1) (Marsh et al., 2013), with its Whole Atmosphere Community Climate Model (WACCM) module for the simulation of the atmosphere, along with the sectional CARMA module (Community Aerosol and Radiation Model for Atmosphere (Toon et al., 1988)) to simulate aerosol microphysics. The sectional scheme distributes particles according to their size over 30 size bins, enabling the evolution of the particle size distribution to be traced in detail with no a priori assumptions on particle size. This model with sectional aerosol was previously used by English et al. (2013) to evaluate aerosol evolution and multi-year impacts from the large magnitude eruptions of Pinatubo 1991 and the $100\times$ larger Toba eruption (74000 years before present). Aerosol impacts from large-magnitude eruptions are substantial but limited by particle growth and sedimentation (with a 20-fold increase in AOD following Toba compared to Pinatubo despite its 100-fold increase in $SO_2$ injection). The globally averaged effective radius reached $0.45\,\mu m$ and $1.9\,\mu m$ after the Pinatubo and Toba eruptions, respectively. English et al. (2013) highlight the need to simulate microphysical processes and advantages of a sectional aerosol representation for a more comprehensive understanding of aerosol evolution following volcanic eruptions. This motivates our study that applies a sectional aerosol microphysics modelling approach to simulate aerosol evolution following a moderate-magnitude eruption.

The aims of our study are: i) to simulate the stratospheric aerosol evolution following the 2009 Sarychev eruption, using a model that explicitly accounts for aerosol microphysical processes using a sectional aerosol scheme. This will deliver the first model simulations of the size-resolved stratospheric aerosol evolution to assess impacts following the Sarychev eruption; ii) compare the model output to balloon-based in-situ measurements of size-resolved aerosol and to satellite observations of aerosol optical depth, including accounting for reported measurement limitations. This will deliver an improved model assessment of the aerosol impact in the 12 months following the Sarychev eruption; and iii) to investigate to what extent co-injection of HCl alongside $SO_2$ may have influenced the subsequent stratospheric aerosol processing and atmospheric chemistry impacts.

## 2    Methods

### 2.1    The CESM1(WACCM)-CARMA model: initialization, set-up and data post-processing

Model simulations were performed using the global Community Earth System Model (CESM1) using its Whole Atmosphere Community Climate module (WACCM) linked to the Community Aerosol and Radiation Model for Atmospheres (CARMA) module, involving the sulfur cycle with a sectional aerosol scheme (English et al., 2011). Land, sea-ice, and rivers were active modules, whereas oceans were data-prescribed. The spatial resolution was a longitude/latitude grid of 144 points by





96, respectively (i.e. approx. 2-degree resolution), and over 88 levels of altitude ranging from the ground to approximately 150 km altitude (with approx. 20 levels in the troposphere). Specified dynamics were used, with a nudging towards MERRA meteorological data (Rienecker et al., 2011) at every time step (30 min) with a weight factor of 0.1 towards the analysis, for temperature and wind fields. The following surface emissions were prescribed in the model. For $SO_2$, $NH_3$, black carbon,

organic carbon, $NO_x$, $CH_4$ and CO emissions, the MACCity data set was used (Granier et al., 2011; Diehl et al., 2012; Lamarque et al., 2010; van der Werf et al., 2006). Anthropogenic $CH_4$ emissions were added from the EDGAR v4.2 database (available at http://edgar.jrc.ec.europa.eu) biogenic CO emissions were added from the MEGAN-MACC database (Sindelarova et al., 2014). OCS was prescribed using data from Kettle et al. (2002). $CH_2O$ was input according to the IPCC RCP8.5 scenario (Riahi et al., 2011), and for $H_2$ the ECCAD-GFED3 database was used (van der Werf et al., 2010). For $CO_2$, $N_2O$,

$CCl_4$, $CF_2ClBr$, $CF_3Br$, $CH_3Br$, $CH_3CCl_3$, $CH_3Cl$, CFC11, CFC113, CFC12 and HCFC22 emissions, lower boundary conditions were prescribed following CCMI/RCP8.5 data.

Simulations were set to start on 1[st] January 2009, using the CESM1(WACCM) initial atmosphere state file at that date. This enabled a six-month model spin-up period before the eruption injection on 15[th] June 2009, after which the simulations were continued for one year, ending on 31[st] May 2010. The Sarychev Peak eruption was simulated by injecting volcanic

$SO_2$ (and HCl) gases into model grid boxes corresponding to the location of the volcano (48°N;153°E), over the duration of 15 June 2009, spread evenly between 11 km and 15 km altitude a.s.l. The model's 2.5° longitude × 1.875° latitude grid resolution means that the volcanic plume is initially too dilute in the model compared to reality. This is nevertheless common methodology, see e.g. Haywood et al. (2010).

A detailed chronology of the Sarychev Peak 2009 eruption can be found in Levin et al. (2010) that identified three explosive

periods: on 12-13 June, repeated explosions occurred, reaching heights ranging from 5 km to 10 km; an isolated, high-altitude explosion occurred on 14 June, reaching 21 km altitude; finally, on 15 June, a series of consecutive explosions reached altitudes ranging between 10 km and 15 km (all times are in UTC). The first eruptive clouds on the 11–14 June period were mainly ash (Rybin et al., 2011). We neglected the minor, low-altitude (inferior to 5 km) explosions reported on 11 and 16 June, and injected $SO_2$ continuously for a 24-hr period on 15 June spread evenly between 11 km and 15 km altitude a.s.l. The timing of

the $SO_2$ emissions is based on $SO_2$ satellite retrievals from IASI (Clarisse et al., 2012; Carn et al., 2016; Carboni et al., 2016), MODIS (MODerate-resolution Imaging Spectroradiometer) (Rybin et al., 2011; Realmuto and Berk, 2016), and OMI (Ozone Monitoring Instrument) (Theys et al., 2015) which all show that the majority of high altitude $SO_2$ was released on the 15[th] (and possibly in the early morning of the 16[th]). Haywood et al. (2010) used a total injection mass of 1.2 Tg $SO_2$, which was the $SO_2$ total mass value retrieved on 16 June with IASI. An update of the $SO_2$ algorithm (Clarisse et al., 2012) found a maximum $SO_2$

mass value of around 0.9 Tg; a value which was confirmed with subsequent updates of that algorithm (Carn et al., 2016). It is also consistent with retrievals from OMI (Theys et al., 2015) and MODIS (Realmuto and Berk, 2016). In contrast, the IASI retrievals reported in Carboni et al. (2016) found that the transient $SO_2$ burden reached only up to 0.6 Tg $SO_2$. We consider though that 0.9 Tg of $SO_2$ is the best estimate for the mass of $SO_2$ injected by Sarychev peak into the UTLS. We did not consider any ash emissions.





In a second simulation, 27 Gg HCl was co-injected alongside the 0.9 Tg of $SO_2$. This initialization follows the recent identification of a localized stratospheric HCl enhancement following the Sarychev eruption (Carn et al., 2016), based on analysis of Microwave Limb Sounder (MLS) satellite observations, reporting a $HCl/SO_2$ mass ratio of around 3%. Since the low vertical resolution of MLS in the lower stratosphere makes it difficult to infer the precise injection altitude of HCl, we

assumed an HCl injection altitude identical to that of $SO_2$. A control run without the volcanic gas injection was also performed, enabling anomalies to be calculated. In the present paper, we will refer to control runs as "volcano-off" simulations, and to runs including the eruption as "volcano-on" simulations.

The CESM1(WACCM) atmospheric chemistry scheme includes a detailed sulfur cycle and key stratospheric nitrogen ($NO_y$), halogenated (i.e. chlorine and bromine) and hydrogenated (in particular $HO_x$ radicals) compounds. The formation and micro-

physics of sulfuric acid aerosol particles simulated by the CARMA module is described in detail in English et al. (2011).

The CARMA module in sectional configuration yields particle concentration across 30 size-bins ranging from approximately 0.68 nm to 3.25 μm in dry diameter. Effective radius is also provided as a direct model output. Post-processing of the model output was used to determine wet particle size distributions, extinctions and optical depth. In each model grid-cell, the wet diameter of each size-bin was calculated using a (hygroscopic growth) parameterisation of $H_2SO_{4(aq)}$ particle volume as

a function of acid weight percentage ($wt\%H_2SO_4$), ambient humidity and temperature following Tabazadeh et al. (1997). Extinctions at 750 nm and 550 nm were calculated by combining the particle concentrations across the sectional size bins with the corresponding wet radii and particle refractive indices following Beyer et al. (1996), using a Mie scattering code at the desired wavelength (van de Hulst and Twersky, 1957). The aerosol extinctions were integrated with altitude over the stratosphere to yield stratospheric aerosol optical depth (SAOD).

**2.2 Balloon-borne in-situ and satellite-based remote sensing observations of aerosol and $SO_2$**

The model $SO_2$ output (from simulations with and without HCl co-injection) is compared to vertical columns of $SO_2$ and total (northern hemispheric) $SO_2$ burden derived from the IASI satellite instrument. The Infrared Atmospheric Sounding Interferometer is an instrument present on board the MetOp-A satellite since the end of 2006. It is a spectrometer measuring infrared light spectra at nadir. Its primary goal is to assess for temperature and water vapour content of the atmosphere, but it can

also be used to retrieve the atmospheric concentrations of various gases, amongst which $SO_2$ (Clarisse et al., 2008; Carboni et al., 2016). IASI provides global coverage twice a day and its footprint ranges from circular (12 km diameter at nadir) to elliptical (up to 20 km by 39 km at the end of the swath). For this comparison we use the IASI retrieval of $SO_2$ by Clarisse et al. (2012). We also compare our results to HadGEM2 model simulations of $SO_2$ and earlier IASI $SO_2$ retrievals reported by Haywood et al. (2010).

Comparisons of the modelled aerosols with in-situ measurements are two-fold. First, we compare the model's output with size-resolved aerosol measurements carried out with the balloon-borne STAC Optical Particle Counter (OPC) instrument over Kiruna, Sweden, on 2, 7, 18 August 2009 and 18 May 2010. STAC (Stratospheric and Tropospheric Aerosol Counter) can be borne under stratospheric balloon gondolas, and can measure low concentrations in aerosols (down to approximately $10^{-4}\,cm^{-3}μm^{-1}$ (Ovarlez and Ovarlez, 1995; Renard et al., 2005, 2010). Particles are classified by their diameters into tune-



able size-bins ranging from a few tenths of micrometre to a few micrometres. The counts in each size bin are normalised by the bin width to yield a size distribution. The uncertainty, defined as the relative standard deviation, is 60% for aerosol concentrations of $10^{-3}$ cm$^{-3}$, 20% for $10^{-2}$ cm$^{-3}$, and 6% for concentrations higher than $10^{-1}$ cm$^{-3}$. STAC was operated successfully on eight different balloon flights throughout the August–September 2009 period over Kiruna, Sweden (68°N; 20°E), as part of

the StraPolÉté campaign (French acronym for Stratosphère Polaire en Été), and also in May 2010, as part of the AEROWAVE project (acronym for AEROsols, WAter Vapour and Electricity). Measurements of the STAC instruments are available online at http://www.pole-ether.fr. During these flights, it was demonstrated that STAC passed through the Sarychev plume (Jégou et al., 2013), as explored further in the present paper. Our comparison focuses on the submicron range between ≈0.3 μm and 1 μm diameter. We have performed an interpolation of the counts from the model's size bins to the STAC size bins (and from

the model pressure levels to the observed pressure of the balloon payload) in order to enable a direct comparison.

Second, we also compare the model's aerosol output with in-situ measurements carried out by the OPC of the University of Wyoming (Deshler et al., 2003), flown on stratospheric balloons launched from Laramie, U.S.A. (41°N; 105°W), on 22 June 2009 and 7 November 2009 (Kravitz et al., 2011). For comparison to the model, total particle number above two diameter threshold sizes are considered here: $d > 20$ nm (condensation nucleii, CN) and $d > 0.5$ μm (N). Uncertainties are 85%, 25%

and 8% for concentrations of $10^{-3}$ cm$^{-3}$, $10^{-2}$ cm$^{-3}$ and $10^{-1}$ cm$^{-3}$ respectively (Deshler et al., 2003). These data are available from ftp://cat.uwyo.edu/pub/permanent/balloon/Aerosol_InSitu_Meas/US_Laramie_41N_105W/. They have been derived by the University of Wyoming as follows: the measurement of N is calculated directly from the OPC instrument. The CN is derived from a condensation nuclei counter co-deployed on the balloon payload.

Model SAOD was compared to that derived from extinction measurements by the OSIRIS aerosol instrument (onboard Odin

satellite). The Optical Spectrograph and InfraRed Imaging System (OSIRIS) is a limb sounder able to provide information on the vertical distribution of atmospheric aerosols (Bourassa et al., 2007, 2008) from the upper troposphere up to the lower mesosphere through the analysis of scattered sunlight. This Canadian instrument has been active since November 2001 on board Swedish satellite Odin (Llewellyn et al., 2004). Its global coverage reaches up to 82° in latitude. Odin evolves on a sun-synchronous orbit, and therefore the availability of OSIRIS's measurements is latitude- and time- dependent. Our analysis

focuses on extinction measurements from OSIRIS version 5.07, available from http://odin-osiris.usask.ca/. Importantly, a novel aspect of our study is that our analysis specifically accounts for instrument errors or limitations as reported by Fromm et al. (2014). Model output data have been degraded accordingly. First, the modelled extinctions have been made to saturate at an upper threshold of $2.5 \times 10^{-3}$ km$^{-1}$; then, extinctions have been only integrated above a certain altitude, dependent on the latitude: a linear variation of this lower limit was assumed, from 0.5 km above the tropopause at the equator up to 5.5 km

above the tropopause at the poles. Further details are given in Results, Section 3.4.



| Date | 15 June | 16 June | 17 June | 18 June | 20 June | 23 June | 25 June | 27 June |
|---|---|---|---|---|---|---|---|---|
| Colocation index, with HCl, in % | 90.34 | 41.29 | 17.61 | 20.26 | 16.60 | 12.38 | 11.36 | 5.36 |
| Colocation index, without HCl, in % | 90.36 | 41.25 | 17.55 | 20.23 | 16.62 | 12.85 | 11.30 | 5.48 |

**Table 1.** Colocation indices quantifying the spatial-amplitude agreement in the volcanic $SO_2$ vertical column densities simulated by CESM1(WACCM) compared to IASI observations over the northern hemisphere, for 1–2 weeks after the eruption (dates corresponding to Fig. 1). See Eq. 1 for details of the computation.

## 3 Results

### 3.1 Spatial and temporal evolution of volcanic $SO_2$ vertical column densities

Fig. 1 shows vertical column densities of $SO_2$ from the CESM1(WACCM) simulation in which both volcanic $SO_2$ and HCl were injected (right-hand panel) and a comparison with IASI retrievals (left-hand panel). Both sets of maps are shown with the same lower threshold in terms of Dobson units, corresponding to an estimated lower threshold of $0.3\,DU$ in IASI's retrievals for this precise eruption and for the IASI retrieval algorithm used (Clarisse et al., 2012). The spatial and temporal evolution of the Sarychev $SO_2$ plume is reasonably well simulated by the CESM1(WACCM) runs throughout the first fortnight following the eruption. There are some notable discrepancies for instance on 16 June 2009 south-west of Alaska: this is because our simulation does not account for the small amount of $SO_2$ that was emitted before the main eruption on 15 June 2009. Also, Asian pollution (close to $0.3\,DU$) is evident in the simulations shown in Fig. 1 but not observed by IASI, likely due to the reduced sensitivity of the IASI retrievals to $SO_2$ below $5\,km$ altitude. To quantify the spatial-amplitude match between the two sets of data, we chose a colocation index calculated as:

$$\rho = \frac{\mathbb{E}[(P_1 - \mu_1)(P_2 - \mu_2)]}{\sigma_1 \sigma_2} \tag{1}$$

where $P_1$ and $P_2$ are the bi-dimensional matrices representing the spatial $SO_2$ loads (for model and satellite retrievals), sampled over the same spatial grid and stacked into monodimensional vectors; $\mu_{\{1;2\}}$ and $\sigma_{\{1;2\}}$ are their respective means and standard deviations. It is expected that the index drops quite quickly after the eruption due to greater dispersion in the model on the $2 \times 2$ degree grids than in the finer-scale (tens of km) IASI observations. Colocation indices were calculated over the first fortnight following the eruption, (Table 1), for the simulation with $SO_2$ injection only and with HCl co-injection alongside $SO_2$. As can be noted from Table 1, co-location indices show comparable values for both model runs. This indicates broadly similar $SO_2$ dispersion in the model runs.

### 3.2 Lifetime, burden of volcanic $SO_2$, and role of co-injected HCl

Fig. 2 shows the modelled northern hemispheric $SO_2$ burden in Tg, calculated by integrating the model anomalies from CESM1(WACCM) simulations with $SO_2$ injection only and with $SO_2$ and HCl co-injection (anomaly denotes a "volcano-on" simulation from which the "volcano-off" control run has been subtracted). Alongside is shown the observed evolution in northern hemispheric $SO_2$ burden derived from the IASI retrieval by Clarisse et al. (2012). The IASI set of data used for this





**Figure 1.** Spatial and temporal evolution of vertical column densities of $SO_2$ (in Dobson Units, DU) over 1–2 weeks following the Sarychev eruption according to IASI satellite observations (left) and simulated by the CESM1(WACCM) model (right). A threshold of 0.3 DU was applied, corresponding to the lower threshold for this precise IASI retrieval (Clarisse et al., 2012). The CESM1(WACCM) model data correspond to instantaneous output at midnight, whereas the IASI data are gathered over the whole of the post-meridiem period.





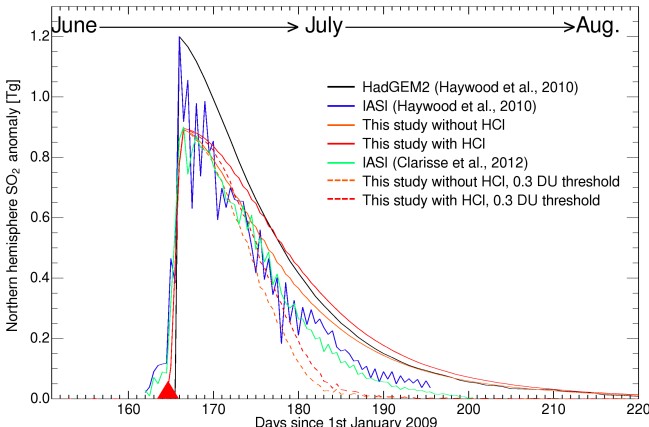

**Figure 2.** Temporal evolution of the $SO_2$ burden (in Tg), integrated over the Northern hemisphere over June–August 2009. Model anomalies are shown for simulations that injected $SO_2$ only (orange) and with co-injection of HCl (red), alongside the IASI retrieval (light blue) with maximum burden of $0.9\,\mathrm{Tg}\,SO_2$. Also shown are adjusted model outputs that account for the $0.3\,\mathrm{DU}$ IASI lower threshold for this particular case (red and orange dashed lines). For comparison, the previously reported model study and IASI retrieval of Haywood et al. (2010) that assumed a higher maximum burden of $1.2\,\mathrm{Tg}\,SO_2$ are also depicted (black and blue lines, respectively).

precise eruption, and considering the retrieval algorithm used (Clarisse et al., 2012), can be considered as showing a lower threshold of around $0.3\,\mathrm{DU}$, therefore two adjusted CESM1(WACCM) model results are also presented that only include data over columns with $> 0.3\,\mathrm{DU}\,SO_2$ to enable a better comparison to the IASI observations. Finally, we also show the northern hemispheric $SO_2$ burden as simulated using the HadGEM2 model (Haywood et al., 2010), and the IASI retrieval reported in

that same study, both of which estimated $1.2\,\mathrm{Tg}\,SO_2$ injection in contrast to the revised IASI analysis (Clarisse et al., 2012) that yielded $0.9\,\mathrm{Tg}\,SO_2$ used in our study.

A notable result is the slower decline in $SO_2$ burden for the model run with volcanic $SO_2$ and HCl co-injection than volcanic $SO_2$ (only). There is also a corresponding slower increase in the sulfate aerosol burden (Fig. 3).

The presence of HCl slows down the oxidation of $SO_2$ to sulfuric acid aerosol particles and hence lengthens the $e$-folding

time of $SO_2$ in the stratosphere by about two days (see calculations below). This is as a result of the competition between their two main oxidation reactions involving OH. These are:

$$HCl + OH \rightarrow Cl + H_2O \tag{R1}$$

and the trimolecular reaction (where M is a third-body, e.g. $N_2$ or $O_2$):

$$SO_2 + OH + M \rightarrow HSO_3 + M \tag{R2}$$

where $HSO_3$ subsequently leads to the formation of $H_2SO_4$ through the reaction sequence described by Weisenstein et al. (1997). This conversion of $SO_2$ to $H_2SO_4$ is limited by the rate of R2 below $40\,\mathrm{km}$ in altitude. Competition between R2 and R1 results in a slower rate of oxidation of volcanic $SO_2$ in the presence of co-injected HCl.





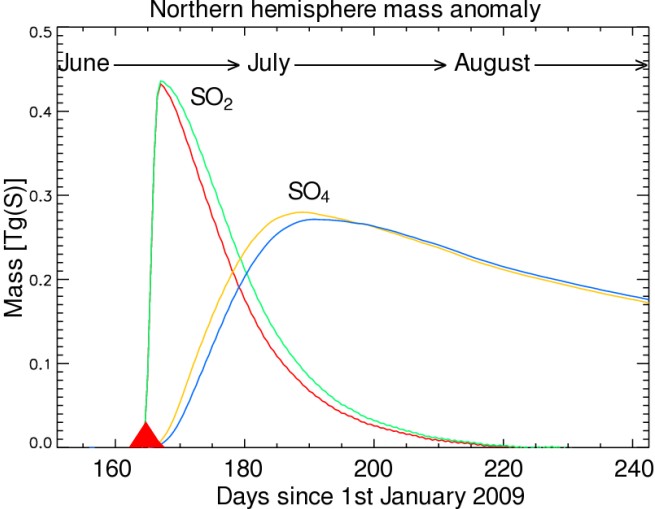

**Figure 3.** Temporal evolution of total $SO_2$ and $SO_4$ burdens (Tg sulfur), integrated over the Northern hemisphere over June–August 2009 (the eruption is depicted by the red triangle). Model anomalies are shown for runs with injection of $SO_2$ only (red and yellow for $SO_2$ and $SO_4$ respectively) and with co-injection of HCl (green and blue for $SO_2$ and $SO_4$ respectively).

A second notable result is that all the (unadjusted) model outputs overestimate the $SO_2$ burden following the eruption compared to IASI measurements. Conversely the two adjusted CESM1(WACCM) model outputs (to correct for the 0.3 DU $SO_2$ lower value for the particular IASI retrievals used) remain in close agreement to the observed post-eruption $SO_2$ burden for the first 1–2 weeks, after which the model-simulated $SO_2$ burden declines more rapidly than the IASI 2012 observations (Clarisse et al., 2012). This evolution can be expected: a greater dispersion in the $2° \times 2°$ model grid cells than in reality (and than observed by the IASI footprint of tens of kilometres), would cause an underestimation of the model $SO_2$ burden compared to IASI. This effect will become more pertinent with dilution over time as the $SO_2$ column approaches the 0.3 DU limit.

In summary, we find that the CESM1(WACCM) model run (adjusted output) with $SO_2$ and HCl co-injection gives best agreement to the IASI $SO_2$ observations. The simulation with $SO_2$ with HCl injection therefore forms the basis for further analysis in Section 3.6. Our model-observation comparison of $SO_2$ burden trends can also be quantified in terms of the $e$-folding time. The definition of the $e$-folding time $\tau$ is the following: let $M(t)$ be the concentration of a species through time; if we assume it follows an exponential decay over a certain period of time $t > t_0$, then $\tau$ is such as $\forall t > t_0, M(t+\tau) = M(t)/e$, id est, $\tau$ corresponds to the time by which the concentration falls to $1/e$ of its initial value. For these calculations we choose the $SO_2$ burden maximum as the initial value. The $e$-folding time-constant for $SO_2$ is approximately 17 days for the simulation including HCl, about two days longer than the approximately 15 days for the simulation that was run without HCl. For comparison, Haywood et al. (2010) report that the HadGEM2 model yields a 13–14-day $SO_2$ $e$-folding time (without the HCl injection, and assuming a higher $SO_2$ injection of 1.2 Tg). Regarding IASI observations, Haywood et al. (2010) report an IASI $SO_2$ $e$-folding time of 10–11 days, whilst using our method we calculate 9 days for the IASI retrieval of 2010. For the IASI





|  | This study, model with HCl | This study, model without HCl | Haywood et al. (2010) HadGEM2 model | Haywood et al. (2010) IASI 2010 | This study, IASI 2010 | This study, IASI 2012 |
|---|---|---|---|---|---|---|
| SO$_2$ $e$-folding time | ≈ 17 days | ≈ 15 days | ≈ 13 ≈ 14 days | N.A. | N.A. | N.A |
| With 0.3 DU threshold | ≈ 11.5 days | ≈ 10 days | N.A. | ≈ 10 ≈ 11 days | ≈ 9 days | ≈ 12 days |

**Table 2.** Comparison of the calculated SO$_2$ $e$-folding times for this study and Haywood et al. (2010). Two sets of IASI data are investigated: 2010 and 2012 retrievals. For the sake of the comparison with the satellite data, a lower threshold of 0.3 DU is applied whenever possible. The most recent IASI data (2012) yield a value close to that calculated with the model simulation of this study with SO$_2$ and HCl co-injection.

SO$_2$ retrieval of Clarisse et al. (2012) we calculate 12 days, i.e. very similar to the adjusted model simulation with SO$_2$ and HCl co-injection (11.5 days). This is summarised in Table 2.

### 3.3 Comparison of the model to in-situ balloon-based measurements of size-resolved aerosol

Here we compare size-resolved aerosol concentrations from our simulations with in situ measurements from balloon-borne
OPCs over Laramie, USA (June, November, 2009) and Kiruna, Sweden (August, September, 2009). It should be emphasized that the instruments are likely to detect a wider range of particles and particle compositions present in the stratosphere, i.e. internally and externally mixed particles with some organic and meteoric component (Murphy et al., 2014), whereas our model simulations provide pure sulfuric acid aerosol particles only. Nevertheless, these are expected to be the dominant source of aerosol in the lower stratosphere in the months following the Sarychev Peak 2009 eruption.

First we compare the model to measurements carried out by the University of Wyoming OPC (Deshler et al., 2003) during balloon-borne flights over Laramie, Wyoming (U.S.A., 41°N, 105°W) on 22 June 2009 and 7 November 2009. These observations were made one week and nearly five months after the Sarychev eruption, respectively. Kravitz et al. (2011) previously suggested that a significant volcanic influence can be seen in the data from 7 November but not on 22 June, based on comparison with balloon flights from other years. Here we compare the data directly to aerosol simulated by our model runs.

Fig. 4 shows both the model and measured aerosol particle number concentrations over Laramie for two particulate size ranges: $d > 20$ nm (noted CN, for condensation nuclei) and $d > 0.5$ μm (noted N). Overall there is good general agreement between simulated and measured values in terms of number concentrations and in the general trend with respect to altitude and size range separation. Note that model-measurement differences are greater in the troposphere since only sulfuric acid particles are simulated.

The upper panel of Fig. 4 (22 June 2009) shows that the volcano-off simulation reproduces the in situ observations of particle number with a very good agreement, supporting the hypothesis of Kravitz et al. (2011) that there was no significant volcanic influence on this day. However, the volcano-on simulation in fact simulates the presence of a volcanic plume, as can be seen by enhancements in CN and N between 13 km and 15 km altitude. We note that the precise geographical location of plume structures is difficult to simulate using low resolution simulations just one week after the eruption. Remote sensing
observations suggest the initial presence of multiple aerosol layers in the stratosphere that subsequently collapsed into a single



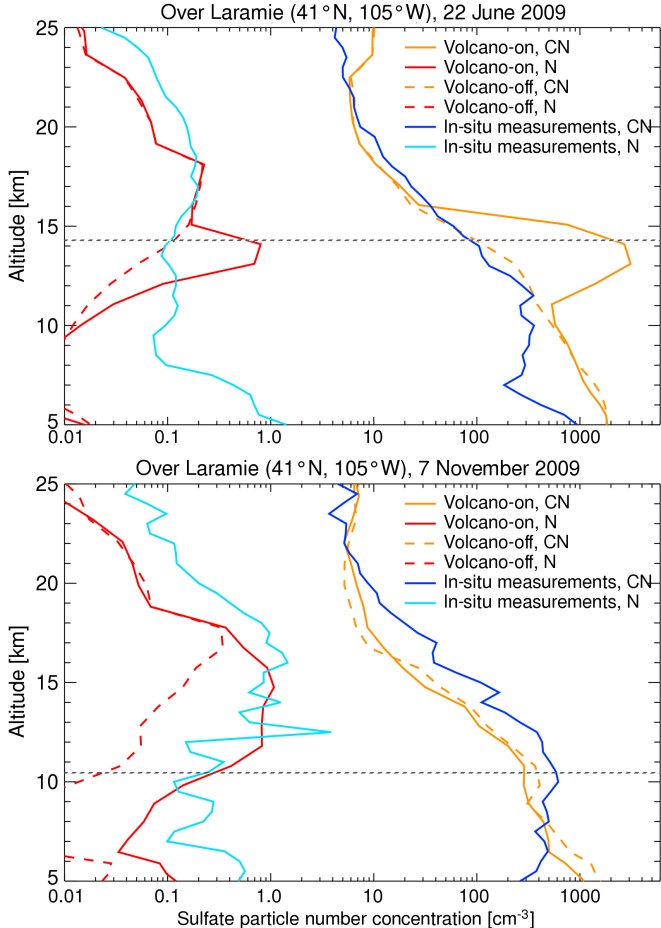

**Figure 4.** Comparison of particle number concentration over Laramie (U.S.A., $41°N$, $105°W$), simulated by the CESM1(WACCM) model (red/orange lines: simulations with and without the Sarychev eruption are shown as full and dashed lines, respectively) and by balloon-borne in situ measurements (blue/cyan lines), for 22 June 2009 and 7 November 2009. Two size ranges are shown: $d > 20$ nm (CN) and $d > 0.5$ µm (N). The model tropopause height is depicted by the dashed grey line. Model uncertainties are greater in the troposphere. On 22 June 2009, the presence of a volcanic plume over Laramie is simulated in the model (evident in both CN and N at the tropopause, discussed in the text), but not evident in the observations. On 7 November 2009 the presence of a more dilute volcanic plume is simulated in the model (evident in N only) that is consistent with the observed N in the lower stratosphere



layer (O'Neill et al., 2012), whereas our CESM1(WACCM) model study assumes injection over 11–15 km. We also suggest that model horizontal resolution effects are a further possible source of error in the volcano-on simulation that might have led to anomalous sulfate plume structure over the measurement location. A geographic 2-D map of the vicinity of Laramie that shows model-simulated sulfuric acid aerosol particles at 13 km altitude, Fig. A1, shows that the location of the measurements lies on

the edge of an aerosol plume structure simulated by the model. Diffusion on the model grids ($2° \times 2°$ resolution) or uncertainties in the initialisation altitude could therefore lead to modelled plume structure over Laramie that is not evident in the observations. Conversely, on 7 November the volcanic plume is simulated to be much more homogeneous (and dilute), covering a larger area that encompasses Laramie. The lower panel of Fig. 4 shows modelled and observed aerosol particle number concentrations for 7 November 2009. For the $d > 0.5$ μm size range (N), the agreement between the volcano-on simulation and the in-situ

measurements below 17 km indicates that volcanic aerosol particles were still present and detectable over Laramie nearly five months after the eruption, and their presence can be quantitatively reproduced by the CESM1(WACCM) model. The profiles from both volcano-on and volcano-off simulations appear very close in the $d > 20$ nm size range (CN) indicating the progressive return to background conditions for this size range.

        Next we compare the CESM1(WACCM) simulations to in situ aerosol measurements made by the STAC instrument on a

balloon gondola in northern Sweden. Fig. 5 compares the particle counts observed by STAC, and the sulfate particle concentrations simulated by the WACCM model for the same location (Kiruna, Sweden, 67°N, 20°E), and times: 2, 7 and 18 August 2009. A comparison is also shown for 18 May 2010 when the stratosphere can be considered to be close to background conditions. The model outputs have been interpolated to the pressure observed by the balloon payload, and to the specific size-bins of the STAC instrument covering 0.325 to 0.885 μm mean diameter.

20       In Fig. 5, a volcanic sulfate aerosol plume can clearly be identified between 11 km and 19 km altitude for all flights in August 2009. This is demonstrated in the third column of the figure by an important difference in modelled particle number over the size-bins of the STAC for the volcano-on and volcano-off simulations: total particle number on the STAC diameter range is enhanced by the volcanic eruption by between one and two orders of magnitude depending on the altitude. Total number simulated in the volcano-on simulation is in good general agreement to the STAC observations.

25       There are some discrepancies between model and observations at higher and lower altitudes: at lower altitudes, the model yields lower counts than the instrument's counts: this is likely due to the presence in the troposphere of non-sulfate aerosols unaccounted for by the model. For the discrepancies above the plume's altitude, the radiometer MicroRADIBAL (French acronym for Micro RADIomètre BALlon) (Brogniez et al., 2003; Renard et al., 2008), flown alongside STAC, identified the presence of some light-absorbing particles around 20 km altitude (Jégou et al., 2013): these might have affected the STAC

measurements (STAC is designed for sulfate particle detection) and were also not included in the model. Their origin is still to be determined. Nevertheless, the good agreement in total number between model and observations in the lower stratosphere (corresponding to the main influence of the volcanic plume) confirms the strong impact of the Sarychev eruption on aerosol number.

        Comparing these aerosol observations above Kiruna in August to those above Laramie in November on an order of magnitude

basis, the Laramie measurements have $\approx 1$ cm$^{-3}$ particulates of diameter greater than 0.5 μm at 14 km altitude in November,





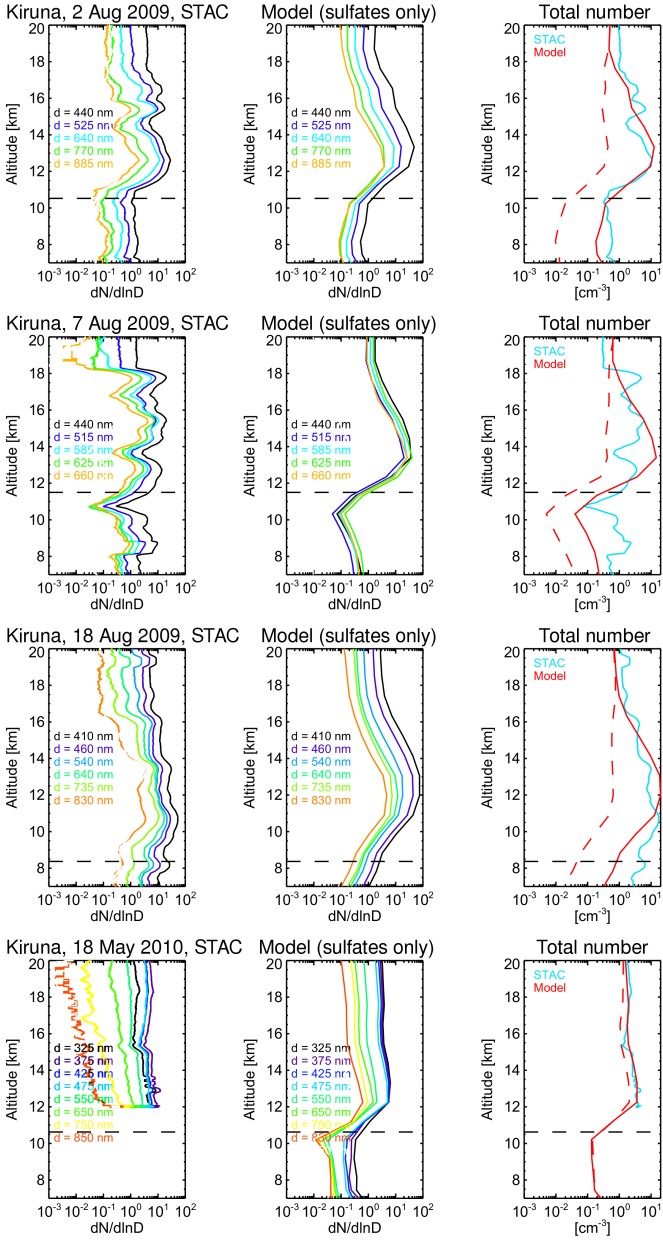

**Figure 5.** Comparison between STAC in situ measurements and CESM1(WACCM) simulations, over Kiruna, for 2, 7 and 18 August 2009 (plume detection) and 18 May 2010 (expected background conditions), and for altitudes ranging from 7 km to 20 km. Left column: particle counts operated by STAC, separated in size bins between 0.325 and 0.885 μm diameter. Second column: simulated equivalent through the use of the CESM1(WACCM) model. Third column: Comparison of the total particle counts for STAC and the model, over the STAC size-range. The red dashed line shows results from the simulation without volcanic aerosols. The model tropopause altitude computed is represented by the horizontal black dashed line.





whereas measurements over Sweden in August of the same year show approximately 10 to 100 times more particles of size greater than $0.4\,\mu m$ in diameter. This indicates the result of coagulation, condensation and sedimentation processes: two months after the eruption there is a strong volcanic impact, but few sub-micrometer size volcanic particles are left in the stratosphere five months after the eruption. The volcanic aerosol evolution is discussed further in Section 3.5 below.

Fig. 6 shows the particle size distributions measured by the STAC, separated in 1-km layers of altitude, for the same four flights as Fig. 5, and compares these to size distributions simulated by the CESM1(WACCM) model. The size distributions are displayed in terms of number, surface and volume, and should be read by pairs, comparing the STAC observations to the control run (volcano-off) on the one hand, and the simulations including the volcano eruption (volcano-on) on the other hand.

The figure highlights that the control run underestimates the particle number (area or volume) size distribution curves by
orders-of-magnitude compared to the STAC observations. A much better agreement is found when the volcanic emission is included in the model simulations, showing a good ability of the model to reproduce volcanic aerosol plume in term of aerosol size distribution. For 18 May 2010, nearly one year after the eruption, the difference between the volcano-on simulation and control run is much less noticeable. This comparison reflects the ability of the model to simulate stratospheric aerosol size distributions in background (or near-background) conditions.

**3.4    Comparison of the model SAOD to OSIRIS observations**

Extinction data from OSIRIS have been used for model and observational assessment of stratospheric aerosol impacts following the Sarychev 2009 eruption (Haywood et al., 2010; Kravitz et al., 2011; O'Neill et al., 2012; Jégou et al., 2013). However, as mentioned in the Introduction, biases in the OSIRIS measurement following volcanic eruptions can affect the reported model-observation comparisons.

In Fromm et al. (2014), a detailed analysis of OSIRIS's limitations was carried out. These authors have shown that two main factors affect the derivation of SAOD by OSIRIS: (i) an upper detection limit on the value of extinctions, above which the measured values saturate; (ii) a latitude dependence in the minimal altitude above which extinctions are integrated to yield the SAOD.

As pointed out by Fromm et al. (2014), it is impossible to revert this process of data degradation; the best we can achieve
to perform consistent model-to-observation comparisons is to degrade the extinctions derived from the model in order to derive SAOD "as OSIRIS would detect it". It must nonetheless be emphasised that such a comparison is not a complete evaluation of the model performance: any agreement found cannot fully validate aspects of the model output that are removed in the degradation process. Nevertheless, such a comparison of the degraded model to (biased) satellite observations is highly valuable: it enables an assessment model performance on a global-scale, which cannot be achieved using local-scale in-situ
observations.

We use the following method: first, we allow the extinctions calculated in the model to saturate, with an upper threshold of $2.5 \times 10^{-3}\,km^{-1}$ corresponding to the detection limit described in Fromm et al. (2014). Second, extinctions are integrated over truncated vertical columns of the atmosphere, introducing a lower altitude limit dependent on the latitude and the local tropopause height. Following Fromm et al. (2014), we define the minimal altitudes $Z_{min}$ above which the extinctions are



**Figure 6.** Particle size distributions in terms of number, area and volume, separated for different altitude layers, shown for the same four days of interest already presented in Fig. 5. Size distributions observed by STAC are shown as solid lines and simulated by CESM1(WACCM) as dashed lines. Graphs go by adjacent pair: comparison of STAC data to both the volcano-off and the volcano-on cases highlights the improved agreement between model and measurements for simulations when the volcano is active.



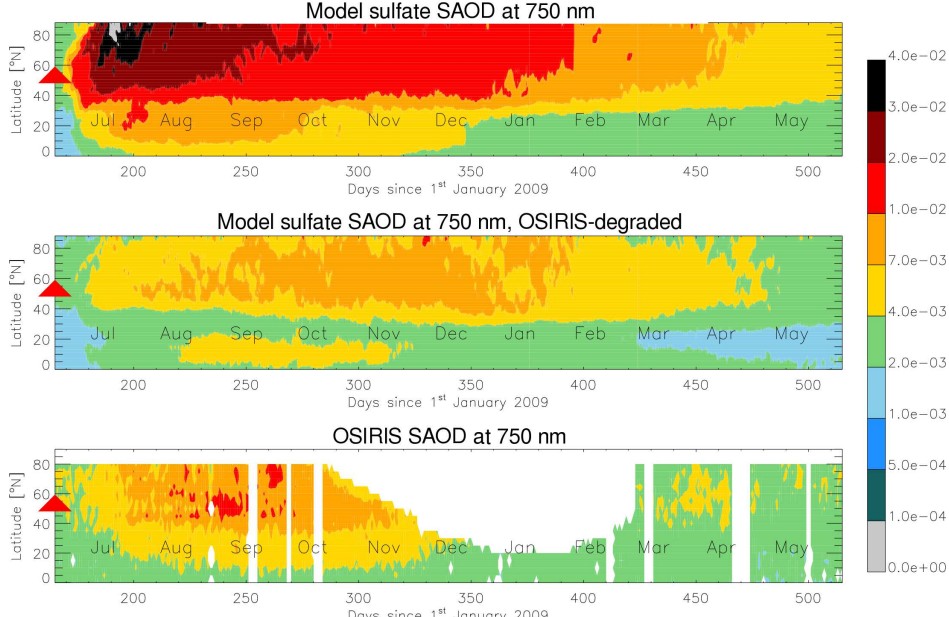

**Figure 7.** Comparison of modelled and observed stratospheric aerosol optical depth. Top panel: stratospheric sulfate aerosol optical depth at 750 nm as simulated by CESM1(WACCM). Middle panel: CESM1(WACCM)'s stratospheric sulfate AOD at 750 nm degraded to account for limitations in OSIRIS data (including saturation effect and minimum altitude). Bottom panel: actual OSIRIS SAOD retrieval obtained from data with measurement limitations. See text for details.

integrated as:

$$Z_{\min}(\lambda, \phi, t) = Z_{\mathrm{trop}}(\lambda, \phi, t) + \Delta(\phi) \qquad (2)$$

where $Z_{\mathrm{trop}}$ is the local tropopause height, $\lambda$ is longitude, $\phi$ latitude, $t$ time, and $\Delta$ is a positive offset function, which was taken in our case as linearly varying with latitude from $0.5\,\mathrm{km}$ at the equator to $5.5\,\mathrm{km}$ at the poles. These were chosen as a

trade-off between the histogram of values (evaluated for 2012) in Fromm et al. (2014), and actual minimum altitudes reached by OSIRIS in 2009. For this series of calculations, dynamical tropopause heights were diagnosed in the model. We verify the broad consistency of these altitude limits for OSIRIS data during the 2009-2010 Sarychev post-eruption period in Fig.A2. Integrating the model-simulated saturated extinctions at $750\,\mathrm{nm}$ over the truncated altitude columns as defined above gives SAOD values that can be considered reasonably consistent with the measurements performed by OSIRIS.

Fig. 7 shows in the top panel the zonally averaged stratospheric sulfate AOD, through time, over the Northern hemisphere, as computed by CESM1(WACCM) in the volcano-on simulation (with co-injection of HCl). A degradation of the model data was then performed following the method described above. The resulting estimation of the sulfate SAOD "as would be detected by OSIRIS" is shown in the middle panel. The bottom panel shows the observed SAOD measured by OSIRIS. Over the winter months there is a lack of observational data from mid-October 2009 until the beginning of 2010, particularly at high latitudes

that coincides with the polar night. A precise comparison for these months is therefore difficult. CESM1(WACCM) suggests



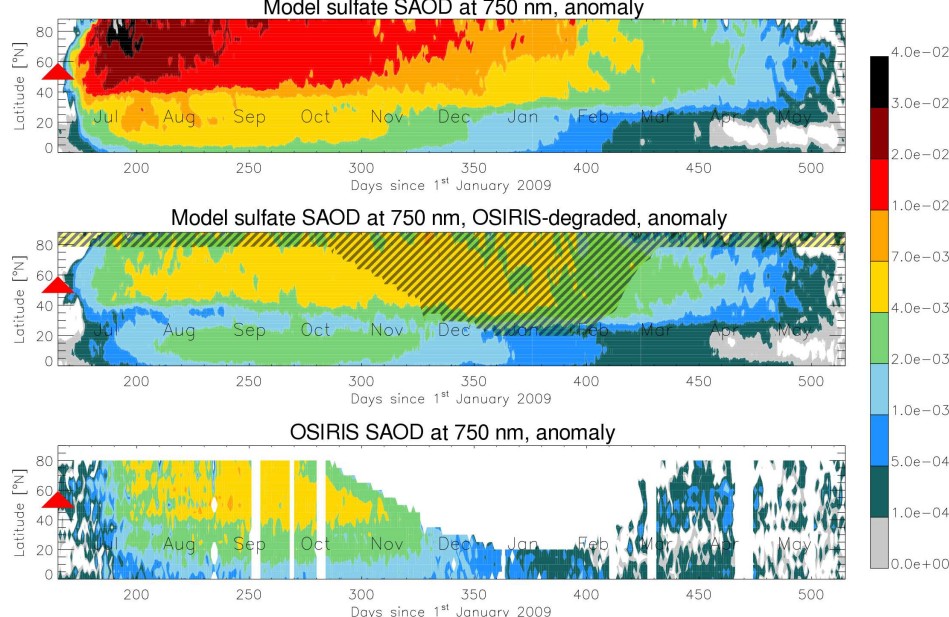

**Figure 8.** Comparison of modelled and observed anomalies in stratospheric aerosol optical depth. The absolute SAOD data from Fig. 7 have been converted to anomalies by subtracting modelled or observed SAOD one week before the eruption. Top panel: stratospheric sulfate aerosol optical depth anomaly at 750 nm as simulated by CESM1(WACCM). Middle panel: CESM1(WACCM)'s stratospheric sulfate SAOD anomaly at 750 nm degraded to account for limitations in OSIRIS data (including saturation effect and minimum altitude). Bottom panel: actual anomaly in OSIRIS SAOD retrieval obtained from data with measurement limitations. The shaded area denotes the polar night, where OSIRIS's measurements are missing. See text for details.

that the sulfate SAOD remains at a fairly constant level over the Northern hemisphere over the October–December 2009 period, then decreases quite quickly from February 2010 to April 2010.

The degraded model SAOD shows reasonable agreement to the SAOD observed by OSIRIS, whilst the non-degraded model simulates much higher SAOD. This demonstrates that OSIRIS's limitations are crucial to the interpretation of its data. In

5 Fig. 7, the observed (OSIRIS) SAOD shows, however, a slightly stronger maximal magnitude than the degraded SAOD from the model. A possible explanation may be that CESM1(WACCM) yields extinctions for sulfuric acid particulates only, whereas OSIRIS's observations account for a more comprehensive SAOD that can include non-sulfate compounds in the lower stratosphere.

To place a greater emphasis on sulfuric acid particulates due to the volcanic eruption, we convert all three datasets to anoma-

10 lies. These anomalies were calculated by subtracting background conditions to the SAOD's, for which averages calculated on the first week of June 2009 were used as an approximate reference. Fig. 8 presents the same layout as Fig. 7, but now displays the SAOD anomalies over the same period (1 June 2009 until 31 May 2010). Again, a good accordance is found between the degraded model compared to OSIRIS (with the non-degraded model showing higher SAOD's). The agreement in SAOD



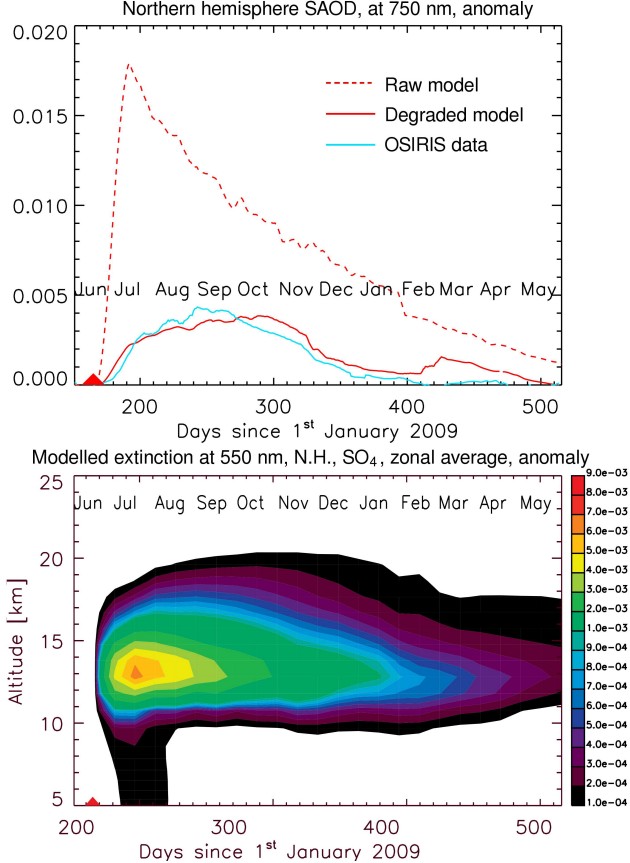

**Figure 9.** (Upper panel) Northern hemisphere SAOD anomalies at 750 nm calculated by integrating the model-simulated extinction (dashed red line), then degraded (full red line), and comparison with OSIRIS's actual data (blue line). The Sarychev eruption is symbolised by the red triangle. (Lower panel) Modelled temporal evolution of the sulfuric acid aerosol extinction coefficient at 550 nm, zonally averaged for the Northern hemisphere, displayed in anomaly (volcano-on minus volcano-off).

anomalies in Fig. 8 is better than for the absolute SAOD's in Fig. 7. This indicates that differences in the background aerosol content prior to the eruption may explain some of the model-measurement discrepancy in term of SAOD maximum amplitude as highlighted in Fig. 7.

Integrating anomaly data presented in Fig. 8 yields the Northern hemisphere SAOD anomaly calculated at 750 nm over the year following the eruption, shown in Fig. 9. The dashed red line is SAOD simulated by the model. The plain red line is the same data after the OSIRIS bias degradation, and the blue line is SAOD from OSIRIS observations. Note that missing data in OSIRIS's measurements during winter was taken into account in the integration of the degraded model data, as shown by the shaded area in Fig. 8. Fig. 8 along with Fig. 9 point out very clearly that taking into account OSIRIS's limitations gives a very good match between simulated and measured AOD values. The bottom panel in Fig. 9 shows the modelled temporal evolution





| Reference | (Haywood et al., 2010) | | Kravitz2011 | | Present study with CESM1(WACCM) | | |
|---|---|---|---|---|---|---|---|
| Latitude band | HadGEM2 | OSIRIS | ModelE | OSIRIS | Raw model | Degraded | OSIRIS |
| 60°N to 80°N | 60 days | 66 days | 57 days | 81 days | 99 days | 45 days | 41 days |
| 40°N to 60°N | 74 days | 75 days | 57 days | 147 days | 105 days | 45 days | 38 days |
| 20°N to 40°N | | | | | 120 days | 49 days | 31 days |
| 0° to 20°N | | | 60 days | 408 days | 65 days | 41 days | 41 days |
| Northern hemisphere | 71 days | 81 days | | | 169 days | 51 days | 52 days |

**Table 3.** SAOD $e$-folding times calculated for the model simulations and for OSIRIS's data reported in previous publications (Haywood et al., 2010; Kravitz et al., 2011) and in the present study. Different latitude bands are considered, and the decay times are all calculated considering SAOD values.

in 550 nm extinction coefficients, zonally averaged for the Northern hemisphere, again highlighting maximum aerosol content around mid-July 2009.

Comparing the direct output of the model to OSIRIS in both Fig. 8 and Fig. 9 highlights a very much stronger and faster formation of sulfuric acid aerosols in the model than can be detected by the OSIRIS instrument, which experiences strongest
measurement-biases shortly after the eruption. Further quantification is given below including $e$-folding times.

Analysing the temporal evolution of the (non-degraded) model SAOD identifies a peak in the Northern hemisphere 750 nm SAOD of $\approx 0.018$ on 12 July 2009 (9), followed by a long decay with an $e$-folding time of $\approx 169$ days. Conversely, OSIRIS shows a much fainter and later peak ($\approx 0.004$ on 1 September 2009), with a quicker decay ($e$-folding time of 52 days). The degraded model SAOD yields an $e$-folding decay time (51 days) that is very comparable to that from OSIRIS; the peak value
is also similar in amplitude to OSIRIS's and is reached on 17 October. This is slightly later than in the OSIRIS data, although the plateau in SAOD during that period can account for this delay.

Table 3 summarises the SAOD $e$-folding times calculated in this study, along with values from previous studies (Haywood et al., 2010; Kravitz et al., 2011), and calculated using OSIRIS data. Different bands of latitude are explored. One can note that the $e$-folding times vary quite significantly between authors, including those computed from OSIRIS's data (it is likely that
different versions of the OSIRIS data—v.5.05 up to v.5.07 for the present study—were used). The main point to be highlighted here is the fair consistency obtained between $e$-folding times computed on the CESM1(WACCM)'s degraded data and on OSIRIS retrievals in our study, as evident from the last two columns of Table 3. We find that both the saturation limit and the fact that extinction profiles may terminate well above the tropopause are significant sources of measurement bias that need to be taken into account in comparison of OSIRIS data to model studies.

## 3.5   Post-eruption effective radius simulated using a sectional aerosol scheme

Discrepancies in the magnitude and $e$-folding times between model and OSIRIS SAOD's have been mentioned elsewhere, notably in Haywood et al. (2010); Kravitz et al. (2011); O'Neill et al. (2012). This led to a consequent questioning of the models' reliability to simulate sulfuric acid particle formation accurately in terms of timing, thought to be caused by the



absence of nucleation of new particles in the model (Haywood et al., 2010; Jégou et al., 2013). Conversely, our study using the CESM1(WACCM) model, whose aerosol microphysics includes nucleation, finds very good agreement with OSIRIS retrievals of SAOD in terms of magnitude and temporally when the model SAOD is degraded to account for both saturation and minimum altitude limitations on the SAOD derived from OSIRIS measurements. The maximum in our (non-degraded) model SAOD is

significantly higher (by a factor $\approx 4.5$) than estimated by both OSIRIS and earlier modelling studies of the 2009 Sarychev Peak eruption (Haywood et al., 2010). A key unconstrained parameter in these earlier studies was the stratospheric particle size distribution that exerts a strong influence on SAOD. It was set to yield an effective radius of around $r_{\mathrm{eff}} = 0.13$–$0.15\,\mu\mathrm{m}$ in Haywood et al. (2010), with the model results from Kravitz et al. (2011) also adjusted to represent this size. Previous studies suggested higher $r_{\mathrm{eff}}$ for large magnitude eruptions that injected $SO_2$ higher into the stratosphere (yielding longer-lived sulfate

clouds): Russell et al. (1993) derived $r_{\mathrm{eff}}$ of $0.22 \pm 0.06\,\mu\mathrm{m}$ around one month after the Mt. Pinatubo 1991 eruption, whilst Stothers (1997, 2001) suggest post-eruption $r_{\mathrm{eff}}$ grew from around 0.2–0.3 to 0.4–0.5 $\mu\mathrm{m}$ over the time-scale of one year. Conversely, a lower $r_{\mathrm{eff}}$ was thought to be reasonable for the moderate-magnitude 2009 Sarychev Peak eruption that injected to the lower stratosphere (yielding relatively fresh and shorter lived sulfate cloud), and appeared consistent with ground-based remote sensing at Mauna Loa (Hawaii, U.S.A.) (Barnes and Hofmann, 2001; Haywood et al., 2010).

Here, the sectional aerosol representation with full aerosol microphysics in CESM1(WACCM) enables to freely simulate the post-eruption evolution in particle size, without any a priori assumptions. Sulfuric acid is first produced by the oxidation of volcanic $SO_2$ which leads to formation of new sulfuric acid particles by nucleation. Processes such as particle coagulation and condensation of sulfuric acid onto the existing particles causes particle growth. Particles are removed from the stratosphere by sedimentation and tropopause folding. The balance between these processes determines the overall size distribution and its

effective radius. Fig. 10 shows the zonally averaged effective radius simulated by the model for three latitude bands (20°N to 40°N, 40°N to 60°N and 60°N to 80°N). Particle growth occurs in regions with elevated sulfate following the volcanic eruption (Fig. A3, supplementary Material). Particle size grows to reach a maximum in zonal mean $r_{\mathrm{eff}}$ of up to $0.2\,\mu\mathrm{m}$ in the lower stratosphere. The greatest enhancement in $r_{\mathrm{eff}}$ occurs at high latitudes as expected given the poleward atmospheric transport in the stratosphere. At mid-latitudes, a temporary decrease in $r_{\mathrm{eff}}$ can also be seen immediately following the eruption:

this is due to new particle formation (nucleation) of particles of a few nm-size. The latitudinal trend in $r_{\mathrm{eff}}$ simulated by our model is broadly consistent with the trend reported from ground-based remote sensing at Eureka (Nunavut, Canada) that found $r_{\mathrm{eff}} = 0.29\,\mu\mathrm{m}$ (O'Neill et al., 2012). Modelled absolute values of $r_{\mathrm{eff}}$ are also globally consistent with balloon-borne observations in August 2009 (Jégou et al., 2013). Aerosol size or $r_{\mathrm{eff}}$ exerts a strong influence on SAOD (e.g. Haywood et al. (2010)). A priori assumptions in stratospheric particle size are a thus major source of uncertainty in model studies that do not

freely simulate the aerosol size-evolution, and that will tend to cause an underestimation of SAOD in cases where the assumed $r_{\mathrm{eff}}$ is lower than reality.

### 3.6 Effects of $SO_2$ and $HCl$ co-injection on stratospheric chemistry

Most studies investigating impacts from modern-day eruptions' stratospheric chemistry have focused on the role of sulfuric acid particles in reducing $NO_x$ levels and activating pre-existing chlorine and bromine in the stratosphere (Fahey et al., 1993;



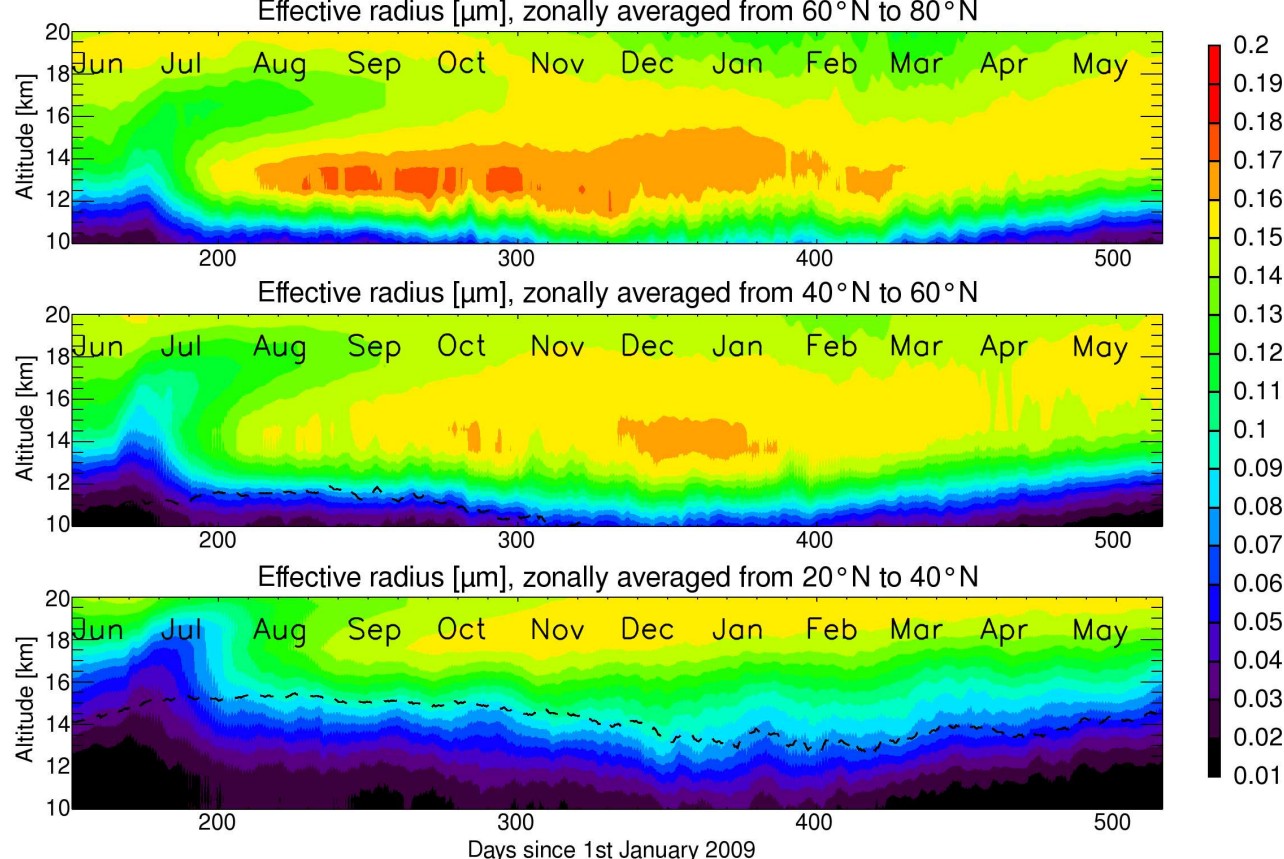

**Figure 10.** Zonally averaged effective radius simulated by CESM1(WACCM) model, in μm , as a function of altitude for three latitude bands (20°N to 40°N, 40°N to 60°N and 60°N to 80°N). The model tropopause is shown as a dashed line.

Solomon, 1999). One must note that halogens from the 1991 Mt. Pinatubo eruption were efficiently washed out and therefore did not reach the stratosphere (Mankin et al., 1992; Tabazadeh and Turco, 1993). Observational evidence of stratospheric $NO_2$ depletion following moderate-magnitude volcanic eruptions is provided by Adams et al. (2017) based on satellite remote sensing, and Berthet et al. (2017) by balloon-borne observations following the Sarychev Peak eruption. Our study builds on these recent works in two aspects: first by using the CESM1(WACCM) model with sectional aerosol representation we freely-simulate the aerosol surface area (SAD) (a function of particle number and size) that is a key control on stratospheric chemistry impacts. Second, we investigate the stratospheric chemistry influence of volcanic HCl that observations show was co-injected alongside $SO_2$ (Carn et al., 2016). The simulated anomalies in ozone and $NO_2$ for the latitudinal bands 40°N to 60°N and 60°N to 80°N are shown for simulations with $SO_2$ injection only, and for HCl co-injection with $SO_2$ in Fig. 11. In summer, greater depletions of up to −60% for $NO_2$ are found at higher latitudes. This is primarily due to the latitudinal extent of the volcanic cloud with higher aerosol loadings in these regions, keeping however in mind that $NO_x$ reduction saturates at





a certain level of SAD (Fahey et al., 1993), and to favored solar illumination conditions for which the catalytic ozone loss cycles (through OH radical production) are enhanced (Berthet et al., 2017). As the season progresses more $NO_x$ is converted to the $N_2O_5$ nitrogen reservoir at nighttime with decreasing solar illumination conditions (especially at high latitudes) and the subsequent conversion to the more stable $HNO_3$ reservoir by enhanced hydrolysis of $N_2O_5$ on volcanic aerosol sequesters

more $NO_x$ at higher latitudes. As the polar vortex builds up, decreasing temperatures in the high latitude stratosphere favor chlorine activation by temperature-dependent heterogeneous processes (mainly involving $ClONO_2$) leading to denoxification (i.e. loss of $NO_x$ by enhanced reaction of $NO_2$ with ClO and by conversion of the $ClONO_2$ nitrogen reservoir to more stable $HNO_3$) and ozone depletion (Solomon, 1999).

Our results are broadly consistent with the observations of Adams et al. (2017) who reported that stratospheric $NO_2$ abun-

dances were reduced by up to $\approx$45–55% over 40–80°N as consequence of the Sarychev Eruption. Berthet et al. (2017) showed maximum $NO_2$ depletion of $\approx$50% in the summertime lower stratosphere above Kiruna (Sweden) following the Sarychev eruption, based on REPROBUS model simulations with SAD prescribed from observations and without HCl injection. They predicted ozone depletion reached up to 4% in the summertime/early fall lowermost stratosphere. Our CESM1(WACCM) simulation with a sectional aerosol scheme and injection of volcanic $SO_2$ (only) finds similar high latitude maximum $NO_2$

reduction ($\approx$ 50%) and maximum ozone depletion (5%). Interestingly, our simulations suggest somewhat greater maximum depletions in the simulation with co-injected volcanic HCl (7% and 60%, for ozone and $NO_2$ respectively) compared to the simulation with $SO_2$ injection only (5% and 50%, for ozone and $NO_2$ respectively). As a result of the enhanced stratospheric HCl budget throughout the season more ozone-depleting chlorine radicals are expected to be formed due to reaction R1 even at mid-latitude conditions, though the impact on ozone appears limited. The impact on summer and fall $NO_2$ is negligible. In

the polar winter, cold temperatures lead to further chlorine activation through heterogeneous processes enhancing some $NO_x$ and ozone reduction. This study highlights the potential for volcanic HCl to supplement and enhance $SO_2$-sulfate impacts on stratospheric chemistry, for eruptions where there is a significant HCl injection to high altitudes. The influence of Sarychev Peak eruption on stratospheric chemistry is nevertheless relatively modest, due to the moderate eruption size, in terms of both the $SO_2$ and HCl injected amounts.

**4 Conclusions**

We have presented a series of simulations carried out with the CESM1(WACCM) model, for the study of stratospheric chemical impacts from the moderate-magnitude 2009 Sarychev eruption. Associated with the CARMA module, the model explicitly simulates the aerosol size evolution using a sectional aerosol scheme (across 30 size-bins), and includes detailed aerosol microphysics. To simulate the eruption, we assumed a 0.9 Tg injection of sulfur dioxide between 11 km and 15 km altitude

over the day of eruption (15 June 2009). We also investigated the impacts of co-injected volcanic HCl.

Through comparison of the model results with satellite (IASI) retrievals of $SO_2$ and in situ measurements of stratospheric aerosols, we were able to assess the model performance, finding good agreement in terms of plume dispersion (Fig. 1) and particle formation rates (Fig. 2, Fig. 3), particle number concentrations as well as particle size distributions before and following





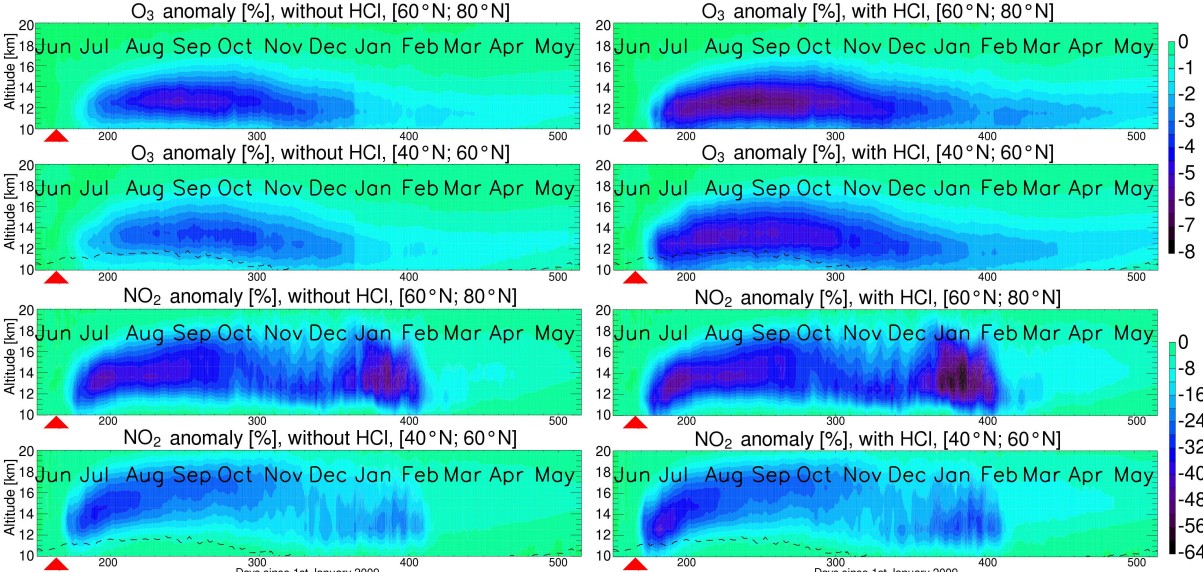

**Figure 11.** Zonally averaged depletions in stratospheric ozone and $NO_2$ at mid- ($40°N$ to $60°N$) and high-latitudes ($60°N$ to $80°N$) following the Sarychev eruption. Ozone is shown in the upper set of plots, $NO_2$ in the lower sets; within each pair, the latitudes are shown as: $60°N$ to $80°N$ (upper plot of pair) and $40°N$ to $60°N$ (lower plot of pair). Simulations by the CESM1(WACCM) model are expressed as percentage anomalies (with respect to the volcano-off control run) and are calculated for the simulation with $SO_2$ injection only (left), and simulation with co-injection of HCl (right). Impacts on stratospheric chemistry are greater at higher latitudes and are enhanced by co-injection of volcanic HCl.

the eruption (Fig. 4, Fig. 5, Fig. 6). In particular, very good agreement was found in terms of particle number concentrations and particle size distributions obtained from balloon-borne observations over Kiruna (Northern Sweden) in July-August 2009 (Fig. 5, Fig. 6) and Laramie (Wyoming, USA) in November 2009 (Fig. 4) confirming the strong impact of the volcanic eruption on the stratospheric aerosol particle load. This suggests that particle formation is represented well in the sectional aerosol

5 scheme (CARMA) in CESM1(WACCM). The simulations suggest that effective radius ($r_{eff}$) becomes enhanced following the eruption to reach up to $0.2\,\mu m$ in the zonal average. This is larger than the fixed aerosol size assumed in previous model studies with limited aerosol microphysics, e.g. $r_{eff} = 0.13$–$0.15\,\mu m$ (Haywood et al., 2010). This overall quantitative agreement lends support to the eruption source parameters used in previous studies: an injection altitude ranging from $11\,km$ to $15\,km$ for $SO_2$, as was already suggested and used in Haywood et al. (2010), appears to be realistic. Likewise, the assumption of a vertical even

10 spread of the total mass of gases injected, and a sole injection of the total gas mass on 15 June 2009, neglecting other minor injections on other days, did provide good results. We also point out that an injected mass of $0.9\,Tg\,SO_2$ (Clarisse et al., 2012; Realmuto and Berk, 2016) instead of $1.2\,Tg$ of previous studies e.g. Haywood et al. (2010) is a fair hypothesis, and enables the model to closely reproduce the observed $SO_2$ burden according to the IASI retrievals of Clarisse et al. (2012).



In addition, we investigated the co-injection of volcanic HCl to the stratosphere. We based our simulations on reported stratospheric HCl/SO$_2$ mass ratio of 0.03 for Sarychev Peak eruption, according to analysis of satellite data by Carn et al. (2016). The altitude and timing of the HCl injection in the model were assumed to be identical to the SO$_2$ injection. Our study suggests that the presence of HCl leads to a delay in the oxidation of SO$_2$ to form sulfuric acid particles of about two days, with a 5–10% increase in the modelled $e$-folding times for SO$_2$. We also find a better temporal accordance in SO$_2$ burden derived from satellite (IASI) data and our simulations when taking HCl into account. The additional surface area provided by volcanic particles catalyses reactions that can perturb stratospheric chemistry, including activation of stratospheric halogens, and can lead to strong reduction of NO$_2$ and modest depletion of ozone as highlighted by Berthet et al. (2017) for Sarychev Peak. Our simulations show that the co-injected volcanic HCl also affects the post-eruption stratospheric chemistry of ozone and NO$_x$, depleting these species more severely than in simulations that account for SO$_2$ injections only. Our results highlight that volcanic HCl emissions should be taken into account when simulating sulfur chemistry and stratospheric chemistry impacts from volcanic eruptions during which HCl is co-injected.

The second major point highlighted by this paper is the treatment of limitations in SAOD derived from OSIRIS measurements: both a saturation effect and a varying minimum altitude in available OSIRIS data (i.e., extinction profiles may terminate well above the tropopause in particular at high latitudes) were identified by Fromm et al. (2014). We used a two-step model degradation process to reproduce these biases in the modelled data, and found as a result very good agreement with the actual OSIRIS measurements following the volcanic eruption, reproducing both the magnitude and temporal evolution of the SAOD following the 2009 Sarychev eruption (Fig. 7, Fig. 8, Fig. 9, Table 3). Recent studies (Haywood et al., 2010; Kravitz et al., 2011; O'Neill et al., 2012) quantifying volcanic impacts have tended to (only) incriminate their models' particle formation schemes because the comparisons with OSIRIS's satellite retrievals were poor specifically regarding the timing of the SAOD maximum. As a matter of fact, caveats on OSIRIS's measurement, as outlined in Fromm et al. (2014), are the key point to any model-observation comparison. We show that there is a considerably improved match between simulated and observed SAODs when these are taken into account. Once again, we stress that this accordance is only obtained by degrading the model output to account for OSIRIS's caveats; the fact that Fig. 8 and Fig. 9 show similar anomaly values cannot be sufficient to thoroughly validate the modelled SAOD's through time. Rather, they provide supportive evidence to our study of stratospheric aerosol evolution following the Sarychev Peak 2009 eruption, using a model with detailed aerosol microphysics and sectional aerosol representation. The non-degraded output from our model shows substantially higher SAOD (maximum of 0.018 at 750 nm) than observed by OSIRIS (0.004), or as reported by previous model studies (with fixed aerosol size, and limited microphysics). Our study therefore highlights that previous modelling studies (involving assumptions on particle size) that reported agreement to (biased) post-eruption estimates of SAOD derived from OSIRIS likely underestimated the climate impact of the 2009 Sarychev Peak eruption

*Acknowledgements.* This work was undertaken as part of WP7 of the VOLTAIRE LabEx (VOLatils — Terre, Atmosphère et Interactions — Ressources et Environnement), convention number ANR–10–LABX–100–01.



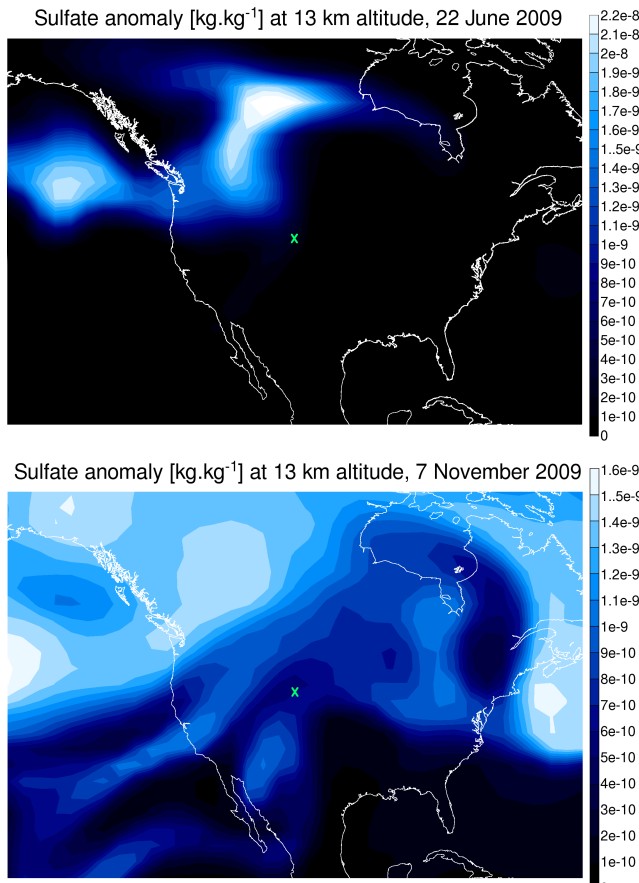

**Figure A1.** Geographic map over the U.S.A. displaying the simulated sulfate aerosol at 13 km altitude on 22 June 2009 (upper panel) and 7 November 2009 (lower panel), as computed by CESM1(WACCM). Note order of magnitude difference in colour scale between the two plots. Laramie is indicated by the green cross: it is located on the very edge of a modelled aerosol plume structure on 22 June 2009, but below a more wide-spread (and dilute) plume on 7 November 2009.





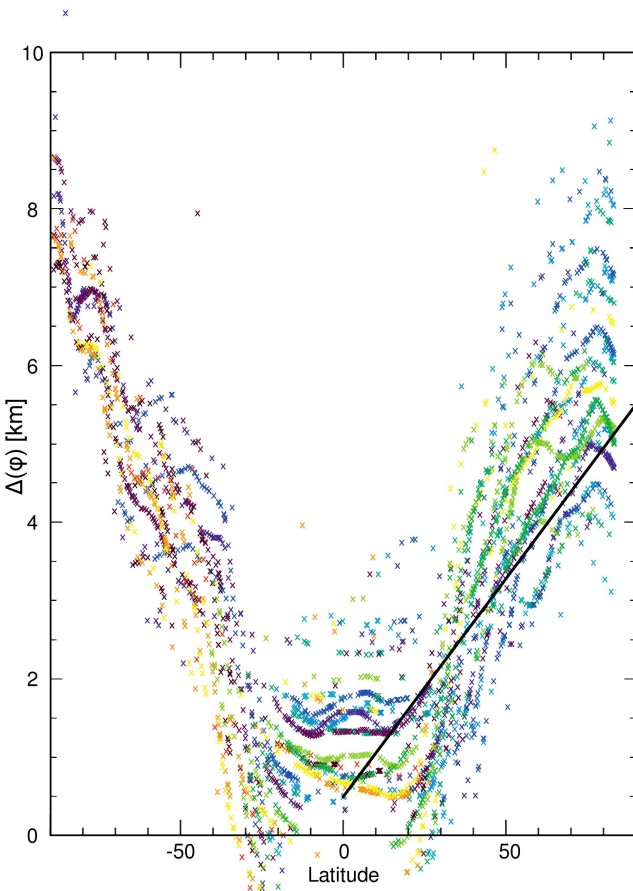

**Figure A2.** Minimum altitude of OSIRIS extinction data calculated relative to the model tropopause as a function of latitude. OSIRIS data are shown for the first of every month from June 2009 to May 2010. There is some missing data during the winter in each hemisphere, particularly at high latitudes. The solid line shows the latitude dependence of the minimum altitude threshold assumed in the model degradation in this study.





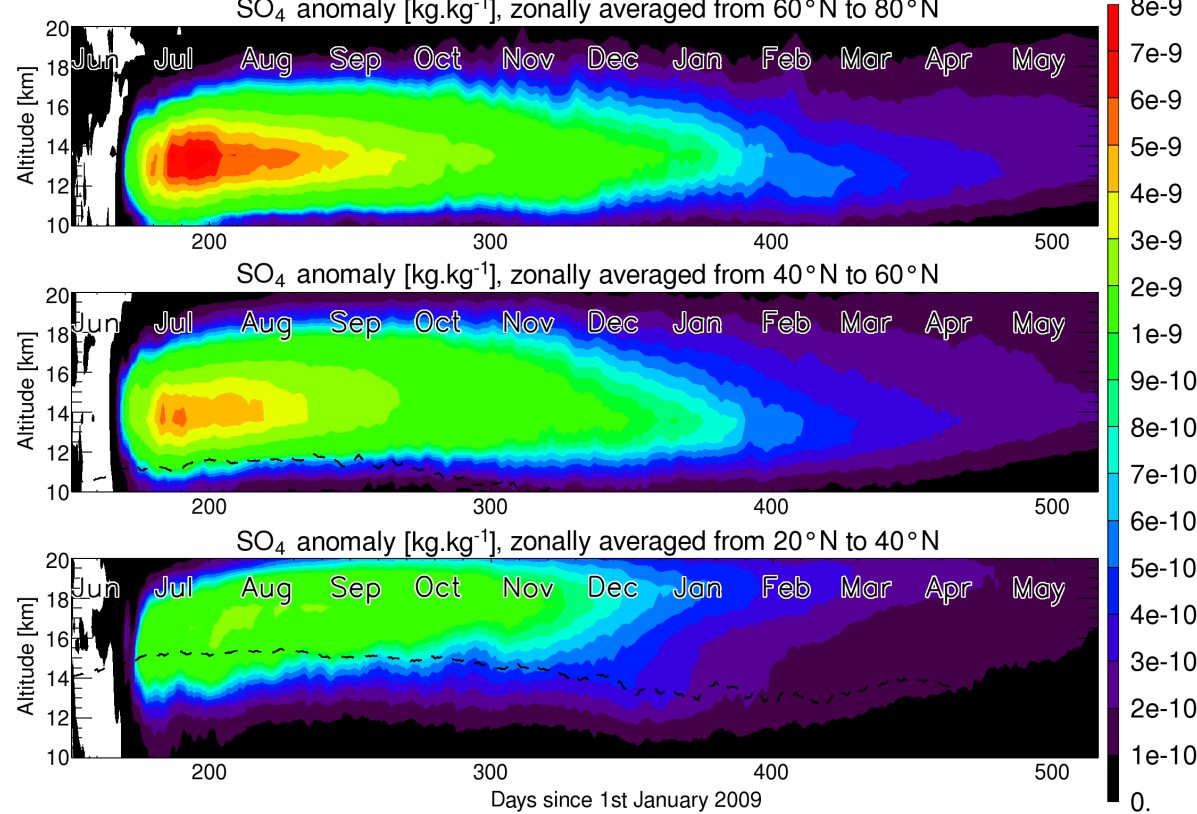

**Figure A3.** Zonally averaged sulfate anomaly, in $\mathrm{kg.kg^{-1}}$, as simulated by CESM1(WACCM) model as a function of altitude for three latitude bands ($20°N$ to $40°N$, $40°N$ to $60°N$ and $60°N$ to $80°N$). The model tropopause is shown as a dashed line.

The authors wish to thank the technical staff of the CaSciModOT structure (Calcul Scientifique et Modélisation Orléans-Tours), part of the French national network of complex systems (RNSC - Réseau National des Systèmes Complexes), along with the CINES (Centre Informatique National de l'Enseignement Supérieur), thanks to which the simulations could be completed. This work was granted access to the HPC resources of CINES under the 2014 allocation (c2014019129) made by GENCI.

5 The authors are grateful to the "Centre National d' Etudes Spatiales" (CNES) balloon launching team for successful operations and to the Swedish Space Corporation at Esrange. The StraPolEté project has been funded by the French "Agence Nationale de la Recherche" (ANR–BLAN08–1–31627), CNES and the "Institut Polaire Paul-Émile Victor" (IPEV). The 2010 balloon observations in the frame of the AEROWAVE project have been supported by the French CNRS-INSU Balloon Committee (CSTB).

The ESPRI-AERIS (formerly ETHER) database (CNES-INSUCNRS) and the CNES "sous-direction ballon" are partners of the project.

10 The authors also would like to thank the NCAR/CESM online discussion board for many helpful technical discussions that helped throughout this study, along with Charles G. Bardeen from NCAR for some discussion about the model, Sophie Bouffiès-Cloché from IPSL for providing the MERRA forcing data, Adam Bourassa from University of Saskatchewan for some discussion about OSIRIS data, and Terry Deshler for providing the Wyoming OPC data.





L. Clarisse is Research Associate (Chercheur Qualifié) with the Belgian F.R.S.-FNRS.

A. Schmidt was supported by an Academic Research Fellowship from the School of Earth and Environment, University of Leeds.



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
