# Peer review of "Model simulations of the chemical and aerosol microphysical evolution of the Sarychev Peak 2009 eruption cloud compared to in-situ and satellite observations"

_Atmospheric Chemistry and Physics, 2017_

## Referee Comment (RC1) · Anonymous Referee #1 · 2 Nov 2017

This is a great study. The importance is clear. The authors have done an excellent job with their analysis - I'm quite impressed with the care and thoroughness they have applied to this research. I only have a few minor comments.

Page 2, lines 24-25: I think you're conflating two issues here. I agree that small eruptions at high latitudes would likely have impacts that are confined to one hemisphere. I also agree that large eruptions in the tropics would likely impact both hemispheres. This is not an either-or. What about small eruptions in the tropics or large eruptions at high latitudes? This sentence needs to be written more carefully.

[Figure]

Page 5, line 17: Did this overly dilute plume affect your results?

Page 5, general: You don't talk too much about the effects of the vertical distribution of the aerosols.

Page 8, line 7: Maybe I'm misinterpreting your colocation metric, but it doesn't look like the plume is "reasonably well simulated" by the model. Some clarity is needed here.

Figure 2 and surrounding analysis: Mills et al. (2016) show that WACCM+MAM3 simulates Pinatubo really well. This is using the same model (albeit with CARMA instead of MAM), but there are some discrepancies in Figure 2. Can you say more about why?

―――――――――――――――

---

## Referee Comment (RC2) · Anonymous Referee #3 · 13 Nov 2017

In this study the authors use CESM1(WACCM)-CARMA simulations to show the impact of volcanic HCl on volcanic $SO_2$ life time and on ozone and $NO_x$ depletion. Further the authors compare their simulations with IASI $SO_2$, balloon-borne particle measurements, and OSIRIS SAOD. Special emphasis was put on the comparison with OSIRIS data accounting for the instrument's limitations.

Fundamentally the study is sound. I recommend it for publication in ACP following revisions suggested below.

[Figure]

**Major comments:**
page 2 line 33: Although the IASI SO2 retrievals are sound and precise, I'm not convinced that they should be the first choice the estimate injection heights. ACE (Doeringer et al, JGR, 2012), CALIOP (e.g. Solomon et al., Science, 2011), MIPAS (Höpfner et al., ACP, 2013, 2015) and the ground based lidar measurements you mentioned clearly show that a significant part of the Sarychev SO2 was injected above 15 km.

page 5 line 14-16: Although you justify your choice of a Sarychev injection on 15 June only into altitudes between 11 to 15 km in the next paragraph there are also studies demonstrating that a substantial amount of SO2 reached higher altitudes (ACE (Doeringer et al, JGR, 2012), CALIOP (e.g. Solomon et al., Science, 2011), MIPAS (Höepfner et al., ACP, 2013, 2015)). Images of different instruments (e.g. http://sacs.aeronomie.be/nrt) show that there was a significant amount of SO2 injected before the 15th and a very recent study in this journal provides an emission time series that placed the onset of the strongest eruption phase in the afternoon of 14 June (Wu et al., ACPD, 2017). Also Levin et al. (2010) found the onset of the second strongest eruptions on 14 June at 18:50. I suggest taking this into account. Please see also further minor comments on this aspect.

**Minor comments:**
page 2 line 15: What do you mean by "global visible AOD was enhanced by up to 0.15"? Is 0.15 a factor or the AOD?

page 2 line 19: Please consider also the Arctic, e.g. Tilmes et al., ACP (2008), as the Sarychev eruption that is discussed here affects the Arctic.

page 2 line 24: Can you add references?

page 2 line 31: Only $SO_2$ and HCl or also ash?

page 3 line 8: Here I'd like to add that a very recent study in this journal found sim-

ulations with a "sedimentation radius" of 0.5–1 $\mu$m to match best with observations (Günther et al., ACPD, 2017).

page 3 line 10-15: The $r_{eff}$ derived from ACE remote sensing measurements was also $0.1-0.3\,\mu$m.

page 5 line 28, 30, 32: What are the uncertainties of the $SO_2$ burdens? Do they agree within their uncertainties?

page 6 line 1: What is the uncertainty of the HCl injection?

page 6 line 18/19: How did you determine the tropopause? What is the uncertainty of the tropopause altitude?

page 6 line 21-29: How do you justify a comparison with $SO_2$ column data, while neglecting all injections below 10 km in the simulations?

page 7 line 6: Please provide a valid URL for the STAC data in indicate your last access (for all urls). After a short search I found the following site claiming to provide STAC data, but ended up at blank pages or 404: http://cds-espri.ipsl.upmc.fr/etherTypo/index.php?id=667L=1

page 8 line 5/11: I suggest considering adding the IASI $SO_2$ retrieval threshold information and its altitude sensitivity range to the description of the data set in Section 2.2.

page 8 line 8: How do you know that this is due to $SO_2$ injected before the 15 June?

page 8 line 11-16: Which model output time did you use for the comparison? The same as the measurement time of each orbit?

page 8 line 24 - page 10 line 6: This part was confusing. I'd suggest reordering and rewording. E.g. present your simulation results first, second your simulation results but with IASI detection threshold, third Haywood model and IASI data. Also consider moving the information on the IASI $SO_2$ retrieval threshold to Section 2.2.

page 9 figure 1: Would your comparison improve if you use 18:00, which is right in the middle of the post-meridiem period, instead of 00:00 model output? What do you mean with "this precise IASI retrieval"? particular? I suggest reducing the number of colors in this figure. I cannot distinguish the many shades of red, blue, and green in the figures. I assume that 7 distinct colors are enough. This type of figure I've just seen in Wu et al. 2017 for a comparison between AIRS data and model output. I suggest a comparison.

page 11 line 1. What do you want to say? Do you mean all model runs or only the "unadjusted" model runs?

page 11 line 2-5. This sentence is confusing. Please clarify.

page 11 line 13/14: Do you mean the maximum on 0.9 Tg here? Please clarify.

page 11 line 15-page 12 line 2: Jumping between your results and the findings of Haywood confused me. Consider presenting your results first and compare then with the results of Haywood.

page 12 table 2: What is the significance of your e-folding time? All results are presented as integers, but the one for your model run with HCl and IASI detection threshold says 11.5 days.

page 12 line 16: Can you quantify the "good general agreement"? Is the agreement in the upper panel of Figure 4 within the error of the OPC and the uncertainty of your volcano-off simulation?

page 13 figure 4: I suggest to add the measurement errors (that are given in Section 2.2) to the OPC data. Without them it is really difficult to judge if the simulations and observations agree quantitatively within their errors on the logarithmic scale. Further, can you indicate the uncertainty range of the simulations?

page 14 line 2: In Section 3.2 (page 10) you mentioned that your model is too disperse. It seems to me that here it is the most likely source of error. I'd suggest to compare with your IASI observations as in Fig. 1 and add this to Figure A1. You could also compare

with Wu et al., ACP, 2017 and discuss.

page 14 line 13: How do you know that the CN particle mode has ever been different from the volcano-off simulation over Laramie?

page 14 line 25-30: You discuss the discrepancies between the measurements and your model results at altitudes below and above your injection height. I assume that not injecting $SO_2$ below 10 km and above 15 km also contributes to the differences. I suggest to add this to the discussion.

page 15 figure 5: Why are you using different colors for similar size bins (e.g. top: 885 nm is orange, bottom, 850 nm is red)? Why are there 3 size bins below 440 nm on 18 May 2010 but the other profiles start with 440 nm? I suggest to merge the lowest size bins for 18 May to make it comparable to the measurements in August. Please indicate the measurement and simulation uncertainty. On a logarithmic scale it's really hard to tell if there is a good agreement. Please also optimize the colors. In the 2 bottom panels there are two indistinguishable green lines.

page 16 line 1: Has there ever been a comparison between STAC and OPC that might explain the difference?

page 16 line 1-3: Does your model simulation suggest coagulation, condensation and sedimentation? What about transport to lower latitudes and dilution between August and November? What is the sedimentation speed and distance of e.g. 0.5 $\mu$m particles over 3 months? Shouldn't they show up at a lower altitude in the OPC data then? Please substantiate your explanation.

page 16 line 32/33: How do you estimate the local tropopause? Do you use the thermal or dynamical tropopause? Which PVU threshold? What do you use at the pole/equator? What is the uncertainty of your tropopause? Please provide details.

page 17 figure 6: As I understood, the main purpose of this figure is to compare the STAC measurements with the model simulations, I suggest to select a smaller range

on the y-axis so that it fits to the STAC data(e.g. dN 1e-3-1e2, dV, 1e-15-1e-12, dV 1e-21-1e-17). In the present figure I can only see that the no-volcano runs do not fit. Further, I consider error bars on the STAC measurements helpful.

page 18 line 4-6: Why don't you rely your $Z_{min}(\Phi, \lambda, t)$ not solely on your analyses of 2009 shown in Fig A2? I consider a 2009 histogram more appropriate than a 2012 histogram with corrections.

page 18 line 6. Ok, it's dynamical tropopause. Which PVU is your tropopause? What do you use in the tropics/at the equator? 380 K? Thermal tropopause? What is the accuracy of your tropopause? Please provide details.

page 28 Figure A2: Please extent the y-Axis to accommodate all data points and provide information on the color code. Does the black line mean that you used only a $Z_{min}(\Phi)$ for your model degradation and not a $Z_{min}(\Phi, \lambda, t)$ as described on page 18? To me it seems that there is some seasonality. Would your analysis improve if you used a $Z_{min}(\Phi, t)$? At high and very low latitudes (0-10N, 50-90N) the minimum altitude threshold seems to be below the median of the data points. What is your reason not using the median?

page 18 figure 7: I suggest checking seasonality for your degradation. Unfortunately I cannot tell from Fig. A2 in which months your degradation altitude fits best, but at high latitudes you have a good agreement in October, November, April, and May and at low latitudes (0-20N) you have the yellow (day 250-350) and blue (day 425-525) features that might coincide with your data points above and below your $Z_{min}(\Phi)$. Please clarify.

page 20 line 1-3: I don't understand what you mean. Please detail where and to what extent the anomalies in Fig 8 agree better than the SAODs in Fig 7. Except from the shaded area indicating OSIRIS measurement gaps in the polar region in the middle panel of Fig. 8 I cannot see obvious additional information. Figure 7 already shows impressively that OSIRIS misses a substantial fraction of lower stratosphere sulfate aerosol.

page 20 figure 9: Why are you showing 550 nm extinction here? It is not used anywhere else, all other OSIRIS data is presented for 750 nm. Please clarify and consider using less colors (7 might be sufficient) in the bottom figure. Some are indistinguishable.

page 21 line 3-5: Please specify what you mean with "strongest measurement-biases shortly after the eruption". Do you mean OSIRIS high Zmin, or its saturation, or its rather coarse sampling that might miss local maxima of the plume filaments shortly after the eruption? Perhaps you want to compare with Günther et al. , ACPD (2017) Fig. 6, which is similar to your Fig. 9, but with different model and satellite data.

page 21 table3, line 12-19: For which purpose do you present e-folding times from other studies? They are not discussed here.

page 22 line 18/19: Please add a reference for those removal processes.

page 22 line 25-28: You might want to include the $r_{eff}$ retrieval by Doeringer et al. (2012), who found 0.1-0.3 $\mu$m for the Sarychev, into your discussion.

page 23 figure 10: Please reduce number of colors. There are too many indistinguishable shades of red and green.

page 22 line 34: Which chlorine and bromine species do you mean?

page 23 line 2: Please note, the washout is not necessarily as efficient as in the Pinatubo case (von Glasow, Chemical Geology, 2009).

page 23 line 20 - page 24 line 2: I did not understand this sentence. Please fix it.

page 24 line 2-5: I did not understand this sentence. Please reword and provide a reference.

page 24 line 5: I suggest starting a new paragraph here to clearly differentiate between heterogeneous reactions on aerosol particles and PSC particles. Isn't HCl the main reservoir of Cl and not ClONO2? There is HNO3 uptake by PSC particles that sediment out and hence lead to denitrification. I suggest to explicitly mention PSCs in

this process and to reword this sentence.

page 24 line 16: is 5% versus 7% a and 50 versus 60% a significant difference in your model? What is the uncertainty?

page 25 line 8: I did not find your results convincing that an injection altitude of 11 to 15 km is realistic. I'd rather interpret your results that there are discrepancies between simulations and measurements above 15 km (see Fig. 4). Further your lower SAOD in the degraded model data compared to OSIRIS (Fig. 7 and 8) may be a result of not accounting for the $SO_2$ injections into altitudes above 15 km that have been observed by several independent measurements.

page 25 line 9-11: Comparing your plume simulation in Fig 1 to IASI data and to the model simulations and AIRS data in Wu et al., ACP (2017), I find some shortcomings in this approach, which become also visible in several details and interpretations on which I commented before. Hence, I suggest to rephrase this sentence and add some discussion on potential errors due to the injection assumption.

page 26 line 29-31: In which respect is this statement different from the findings in Ridley et al. (2014)?

**Technical Suggestions**

page 1, line 19: confirm

page 5 line 5: please sort references chronologically

page 5 line 7, 8: references for Sindelarova and Kettle are missing

page 6 line 22: just write IASI

page 6 line 31: Define abbreviation at first usage only. Please also check in other places e.g. for OSIRIS page 7 line 20.

page 7 line 5: What does StraPolEte stand for? Is there any reference to the "AEROWAVE" project?

page 10 figure 2: In the figure the IASI retrieval is green, but the caption says light blue.

page 11 line 10: "... Section 3.6." Please start a new paragraph here.

page 13 figure 4 caption: "solid" instead of "full" line

page 14 line 7: "... observations." Please start a new paragraph here.

page 14 line 12: Do the profiles only appear to be close to each other or are they close?

page 18 line 15: Do you mean: A comparison for these months is therefore impossible?

page 21 line 7: (9) = (Fig. 9)?

page 21 line 21/22: Please replace "... elsewhere, notably ..." by e.g. and refer to Table 3.

page 22 line 17: ... , which

page 22 line 22: material

page 22 line 29: ...are thus a major ...

page 22 line 33: "... investigating the impacts of modern day eruptions on stratospheric ..." Please fix this sentence.

page 24 line 30: Please write "... over one day of eruption ..."

page 25 line 5: Please write "... suggest that the effective radius becomes ..."
* * *

---

## Author Comment (AC1) · 20 Jan 2018

Interactive comments on

**Model simulations of the chemical and aerosol microphysical evolution of the Sarychev Peak 2009 eruption cloud compared to in-situ and satellite observations**

by Thibaut Lurton et al.

Authors' foreword:

We wish to thank both referees for their comments on the manuscript.

Please find below our answers to their remarks: their original comments are typed in italics, and we address our answers following each point raised.

Answers to Anonymous Referee #1

*This is a great study. The importance is clear. The authors have done an excellent job with their analysis - I'm quite impressed with the care and thoroughness they have applied to this research. I only have a few minor comments.*

*Page 2, lines 24-25*

*I think you're conflating two issues here. I agree that small eruptions at high latitudes would likely have impacts that are confined to one hemisphere. I also agree that large eruptions in the tropics would likely impact both hemispheres. This is not an either-or. What about small eruptions in the tropics or large eruptions at high latitudes? This sentence needs to be written more carefully.*

We agree with the reviewer, as we forgot to mention small tropical eruptions (e.g. Soufrière in 2006 or Kelud in 2014) which affected at least one of the hemispheres, depending on the QBO phase, and likewise, we do not mention major eruptions having affected one hemisphere only. We therefore rewrote the concerned lines:

"However, they typically have a much-reduced effect on climate and atmospheric chemistry compared to large-magnitude eruptions (Oman et al., 2005; Kravitz et al., 2010). In general a smaller mass of $SO_2$ is injected and oxidized to sulfate aerosol. Also, by injecting to lower altitudes, the emissions from moderate-magnitude eruptions are more susceptible to removal by stratospheric-tropospheric exchange processes. Nevertheless..."

*Page 5, line 17*

*Did this overly dilute plume affect your results?*

The question is about the initial dilution of the injected $SO_2$ into the model grid, which is an unavoidable consequence of global model simulations using large grids (few degrees) as for other studies, e.g. Haywood et al. (2010). Nevertheless, we find good agreement in our model-observation comparisons of aerosol several months after the eruption. This indicates that the initial dilution does not seem to have important impact on our comparisons over those time-spatial scales. To fully test the reviewer's question would require modelling over a finer grid. That is beyond the scope of our study, but is indeed of interest for future studies as we can anticipate availability of more powerful computational resources as well as new upcoming satellite data at higher resolution.

*Page 5, general*

*You don't talk too much about the effects of the vertical distribution of the aerosols.*

We have added a sentence to line 18:

"The vertical distribution of our $SO_2$ injection follows previous model studies (e.g. [Haywood, 2010]). It is a somewhat coarse approximation given that |Prata, 2017] report lidar observations of fine-scale aerosol layers shortly after the eruption. Nevertheless, these were subsequently observed to collapse into a single layer in the lower stratosphere. For the magnitude of the $SO_2$ injection we use a revised estimate that contrasts to previous studies, as discussed below."

*Page 8, line 7*

*Maybe I'm misinterpreting your colocation metric, but it doesn't look like the plume is "reasonably well simulated" by the model. Some clarity is needed here.*

The colocation metric we used is as a matter of fact the Pearson's correlation coefficient, which takes into account both differences in amplitude and location. Another alternative would have been to use the Spearman's rank correlation coefficient, which operates on ranked variables, and therefore is less sensitive to strong outliers and does not depend on a linear relationship between the variables. Though this latter coefficient is more flattering in terms of matching scores, we chose to retain our first approach.

We consider our plume reasonably well simulated by the model, as compared visually (Figure 1) to figures from previous studies such as Haywood et al. (2010) (they do not report colocation metrics to compare quantitatively). This is further confirmed by our analysis of Northern Hemisphere SO2 burden in Figure 2.

*Figure 2 and surrounding analysis*

*Mills et al. (2016) show that WACCM+MAM3 simulates Pinatubo really well. This is using the same model (albeit with CARMA instead of MAM), but there are some discrepancies in Figure 2. Can you say more about why?*

The question raised by the reviewer is tricky to address. It is difficult to compare different model configurations and associated simulations conducted for two different eruption events which

correspond to different dynamical contexts (i.e. latitude and altitude range of injection) and aerosol microphysical properties (i.e. size distributions). Perhaps the size distribution shape corresponding to the Pinatubo aerosol are sufficiently addressed with a modal aerosol scheme. However, our Fig. 2, displaying $SO_2$ temporal evolution is not directly comparable to Fig. 5 from Mills et al., which presents AOD. Also, one should note that conversely to our Fig. 2, Fig. 5 presented by Mills et al. is in log-scale so it is somewhat complicated to definitely infer a very good agreement between MAM3 and observations in this study.

*Answers to Anonymous Referee #3*

*In this study the authors use CESM1(WACCM)-CARMA simulations to show the impact of volcanic HCl on volcanic SO2 life time and on ozone and NO$_x$ depletion. Further the authors compare their simulations with IASI SO$_2$, balloon-borne particle measurements, and OSIRIS SAOD. Special emphasis was put on the comparison with OSIRIS data accounting for the instrument's limitations.*

*Fundamentally the study is sound. I recommend it for publication in ACP following revisions suggested below.*

**Major comments:**

*page 2 line 33:*

*Although the IASI SO2 retrievals are sound and precise, I'm not convinced that they should be the first choice the estimate injection heights. ACE (Doeringer et al, JGR, 2012), CALIOP (e.g. Solomon et al., Science, 2011), MIPAS (Höpfner et al., ACP, 2013, 2015) and the ground based lidar measurements you mentioned clearly show that a significant part of the Sarychev SO2 was injected above 15 km.*

*page 5 line 14-16:*

*Although you justify your choice of a Sarychev injection on 15 June only into altitudes between 11 to 15 km in the next paragraph there are also studies demonstrating that a substantial amount of SO2 reached higher altitudes (ACE (Doeringer et al, JGR, 2012), CALIOP (e.g. Solomon et al., Science, 2011), MIPAS (Höepfner et al., ACP, 2013, 2015)). Images of different instruments (e.g.*

*http://sacs.aeronomie.be/nrt) show that there was a significant amount of SO2 injected before the 15th and a very recent study in this journal provides an emission time series that placed the onset of the strongest eruption phase in the afternoon of 14 June (Wu et al., ACPD, 2017). Also Levin et al. (2010) found the onset of the second strongest eruptions on 14 June at 18:50. I suggest taking this into account. Please see also further minor comments on this aspect.*

IASI altitude retrievals have been shown to be accurate in general within 1—2 km (e.g. Clarisse et al. (2014), Carboni et al. (2016)), so that we believe that IASI can be used to make statements on the injected altitude. For the early Sarychev plume, some $SO_2$ is measured up to 20 km, but the

majority (> 95%) of the SO$_2$ mass was found to be below 15 km. This last statement is true for both the retrievals presented in Carn et al., 2016 and Carboni et al., 2016, which each use an independent altitude retrieval algorithm.

Other instruments (limb/occultation/lidar) have indeed a more inherent sensitivity to altitude, although limb/occultation measurements also have limitations in their vertical resolution. However, we disagree that these measurements unambiguously show that a "significant part" of the plume is located above 15 km in the early plume. Part of the problem is that some of these instruments have a very limited coverage, and that therefore measurements are often reported within the aged air masses, which can undergo significant vertical transport over time (Vernier et al., 2011), and thus no longer provide information on the injection altitude.

Here we summarize the observational evidence from the papers that the reviewer mentions:

MIPAS: Höpfner et al. give the following numbers 888 kT (10—14 km) / 542 kT (14—18 km) / 44 kT (18—22 km). It is unclear over which time period these measurements were gathered.

CALIPSO measurements of aerosols: Solomon et al, 2011, Fig. 1, show the SR@532nm between 17 and 21 km, and indeed shows enhancement in the wake of the Saychev eruption. It is however not possible to conclude from this plot that a significant part of SO$_2$ was injected at those altitudes.

Prata et al., ACP, 2017 (part 4.2) indicate that no CALIOP data is available for the 12—14 June period.

Doeringer et al., 2012, Fig. 6, shows a peak in the median atmospheric extinction in July 2009 at 13 km, which quickly drops off below 9 km and above 15km. Figure 7 shows similar result, with a SO2 tail at 16,17 km altitude on 14 July. Fig. 8 shows a similar profile for SO$_2$.

A trajectory-model approach by Wu et al. (2017) finds the largest SO$_2$ injection occurred between 12 and 17 km [Günther et al., 2017].

We must also emphasise on the fact that our model vertical resolution as far as injection is concerned is 1 km. Therefore, the precision in the injection altitudes is somewhat coarse, and subject to some trade-off. A detailed study of the complexities of the injection, tracing the horizontal and vertical transport of fine-scale plume filaments is not the goal of our study at the resolution of our global model (~2 degree grid, ~1 km vertical resolution) that focuses on a detailed chemistry and aerosol microphysics. Such efforts are rather suited to trajectory-dispersion models e.g. the recent study by Wu et al., ACP (2017) of transport pathway of the Sarychev SO$_2$ emission and sulfate aerosol from the extratropical lower stratosphere to the tropical tropopause layer (TTL).

A visual analysis of Figure 2 of Wu et al. (2017) depicts some injection > 15 km in afternoon-evening of 14 June. To estimate the load requires looking at the concentration multiplied by the area (in altitude-time axes) of the plot. This suggests that between 12 and 15 km altitude is a suitable approximation for the majority of the emission, as needed to investigate large-scale evolution patterns. The studies of Wu et al. and Gűnther et al. (2017) indicate that emissions at higher altitude could particularly affect transport of plume to southern latitudes. Therefore our choice of injection altitude may limit our modelling of this aspect, but it is not the focus of our study that focuses on the NH evolution and observations at mid and high latitudes.

On the comment of the eruption time: IASI measures about 100 kT on the evening orbit of the 14[th].

So indeed there was a non-negligible amount of $SO_2$ emitted before the 15$^{th}$, but according to IASI most of it was clearly erupted on the 15$^{th}$ / early 16$^{th}$.

Last, we point out the fact that our study had to keep a relative consistency with the previous studies to which it is compared throughout. This advocates for the injection altitudes and times we chose, which are comparable to those of the papers we use as points of comparison.

***Minor comments:***

*page 2 line 15:*

*What do you mean by "global visible AOD was enhanced by up to 0.15"? Is 0.15 a factor or the AOD?*

It is the AOD. We replaced the text by "global AOD (in the visible) was enhanced, reaching up to 0.15" in the manuscript.

*page 2 line 19:*

*Please consider also the Arctic, e.g. Tilmes et al., ACP (2008), as the Sarychev eruption that is discussed here affects the Arctic.*

We replaced the clause "Antarctic ozone hole" by "polar ozone holes", and made reference to Tilmes et al., 2008.

*page 2 line 24:*

*Can you add references?*

This part of the text was rewritten following Referee #1's remark.

*page 2 line 31: Only SO2 and HCl or also ash?*

Indeed, some ash was injected, though it was not prescribed in our model runs. Mention of ash has been however added to the concerned sentence. We discuss later that we do not consider ash in this study. Here we are focused on $SO_2$ (and co-injected HCl mentioned as it is an aspect of our study).

*page 3 line 8: Here I'd like to add that a very recent study in this journal found simulations with a "sedimentation radius" of 0.5–1 μm to match best with observations (Günther et al., ACPD, 2017).*

We added the reference within the manuscript.

*page 3 line 10-15:*

*The reff derived from ACE remote sensing measurements was also 0.1 – 0.3 μm.*

We have added the sentence:

"Further evidence for a larger particle size comes from effective radius estimate of 0.1-0.3 μm derived from satellite-based observations on month after the eruption (Doeringer et al., 2012) and a particle "sedimentation radius" of 0.5 – 1 μm from a model sensitivity study (Günther et al., ACPD, 2017)."

*page 5 line 28, 30, 32:*

*What are the uncertainties of the $SO_2$ burdens? Do they agree within their uncertainties?*

A typical uncertainty on SO2 burden retrievals using the [Clarisse et al., 2012] algorithm would be 10—20 %. Considering the highest uncertainty, the 0.9 Tg estimation still stands out as being different from 1.2 Tg. This was added to the manuscript (see remark concerning p.8, l.5—11).

*page 6 line 1:*

*What is the uncertainty of the HCl injection?*

Carn et al., 2016 mention HCl at 7—9 ppbv, compared to 529 ppbv $SO_2$, but do not provide any other figure to derive uncertainty, that is difficult to quantify as discussed in their text. As far as our study goes, we tested the sensitivity to the presence of HCl, and therefore our results should be considered as a very first investigation of the impacts of the co-injection of HCl with $SO_2$ from the Sarychev eruption, using the best-available estimate from Carn et al. (2016).

*page 6 line 18/19:*

*How did you determine the tropopause? What is the uncertainty of the tropopause altitude?*

The model tropopause was used (see further details below). Therefore we cannot properly state a tropopause altitude uncertainty; the tropopause height is self-consistent with the model.

*page 6 line 21-29:*

*How do you justify a comparison with $SO_2$ column data, while neglecting all injections below 10 km in the simulations?*

Please refer to the answers to the major comments concerning injection heights. We have no precise number concerning injection below 10 km. Furthermore, as the plume eventually goes down in altitude, it is useful to monitor the column data. We here focus on volcanic sulfuric acid particles in the stratosphere. The tropospheric lifetimes of sulfate aerosols and $SO_2$ are shorter, due to wash-out processes and clouds.

*page 7 line 6:*

*Please provide a valid URL for the STAC data in indicate your last access (for all urls). After a short search I found the following site claiming to provide STAC data, but ended up at blank pages or 404:* http://cds-espri.ipsl.upmc.fr/etherTypo/index.php?id=667L=1

The problem was acknowledged, and the ESPRI website team was notified accordingly.

*page 8 line 5/11:*

*I suggest considering adding the IASI SO2 retrieval threshold information and its altitude sensitivity range to the description of the data set in Section 2.2.*

We added the following sentence to Section 2.2:

"IASI retrievals have a typical altitude sensitivity of 1—2 % km. For the precise Sarychev eruption retrievals, SO2 loads can be expected to have a 10—20 % uncertainty."

*page 8 line 8: How do you know that this is due to SO2 injected before the 15 June?*

We toned down our assertion, rewriting the sentence as follows: "this is likely to be due to our simulation not accounting for the small amount of SO2 that was emitted before the main eruption".

*page 8 line 11-16:*

*Which model output time did you use for the comparison? The same as the measurement time of each orbit?*

No average was performed, in order not to over-dilute the signal. We considered two instantaneous outputs per day, at 0:00 and 12:00.

*page 8 line 24 - page 10 line 6:*

*This part was confusing. I'd suggest reordering and rewording. E.g. present your simulation results first, second your simulation results but with IASI detection threshold, third Haywood model and IASI data. Also consider moving the information on the IASI SO2 retrieval threshold to Section 2.2.*

The text is rearranged as follows:

"Fig. 2 shows the modelled northern hemispheric $SO_2$ burden in Tg, calculated by integrating the model anomalies from CESM1(WACCM) simulations with $SO_2$ injection only and with $SO_2$ and HCl co-injection (anomaly denotes a "volcano-on" simulation from which the "volcano-off" control run has been subtracted). Two adjusted CESM1(WACCM) model results are also presented that only include data over columns with $> 0.3$ DU $SO_2$ to enable a better comparison to the IASI observations. Alongside is shown the observed evolution in northern hemispheric $SO_2$ burden

derived from the IASI retrieval by Clarisse et al. (2012) (that has a lower threshold of around 0.3 DU, see Methods 2.2). Finally, we also show the northern hemispheric $SO_2$ burden as simulated using the HadGEM2 model (Haywood et al., 2010), and the IASI retrieval reported in that same study, both of which estimated 1.2 Tg $SO_2$ injection in contrast to the revised IASI analysis (Clarisse et al., 2012) that yielded 0.9 Tg $SO_2$ used in our study."

We add the following sentence to page 6, line 28:

"For this comparison we use the IASI retrieval of $SO_2$ by Clarisse etal. (2012). The IASI dataset and retrieval algorithm used for this precise eruption can be considered as showing a lower threshold of around 0.3 DU."

*page 9 figure 1:*

*Would your comparison improve if you use 18:00, which is right in the middle of the post-meridiem period, instead of 00:00 model output? What do you mean with "this precise IASI retrieval"? particular? I suggest reducing the number of colors in this figure. I cannot distinguish the many shades of red, blue, and green in the figures. I assume that 7 distinct colors are enough. This type of figure I've just seen in Wu et al. 2017 for a comparison between AIRS data and model output. I suggest a comparison.*

The IASI data are averaged over the AM and PM local periods. We use instantaneous snapshots of the model as a comparison, at 0:00 and 12:00, which is (as the referee points out) not a perfect match, but in our opinion remains sufficient to demonstrate the good behaviour of the model in comparison to the observations.

The clause "this precise IASI retrieval" refers to the possible different computation methods in the IASI retrievals, which can vary. As for the number of colours, we choose not to change it in order to keep a useful high dynamic range on the onset of the eruption.

The comparison with Wu, 2017 indeed shows good agreement. We add, to conclude paragraph 3.1: "The spatial and temporal evolution of the plume in our study is consistent with the results of Wu, et al. (2017), where AIRS data are presented along with simulations by a particle dispersion model."

*page 11 line 1.*

*What do you want to say? Do you mean all model runs or only the "unadjusted" model runs?*

We mean all the unadjusted model runs. This was clarified in the text.

*page 11 line 2-5.*

*This sentence is confusing. Please clarify.*

We propose the following modification to the text:

"A second notable result is that all the unadjusted model outputs overestimate the $SO_2$ burden

following the eruption compared to IASI measurements. The HadGEM2 model $SO_2$ exceeds the Haywood et al. (2010) IASI observations for the whole period. The unadjusted CESM1(WACCM) outputs also exceed the Clarisse et al. (2012) IASI observations after a few days. This behaviour contrasts with the two adjusted CESM1(WACCM) model outputs that correct for the 0.3 DU $SO_2$ lower value of the particular IASI retrievals used. The adjusted CESM1(WACCM) model outputs remain in close agreement to the observed post-eruption SO2 burden for the first 1–2 weeks, after which the model-simulated $SO_2$ burdens decline more rapidly than the IASI 2012 observations. This evolution can be expected: a greater dispersion in the 2°×2° model grid cells than in reality (and than observed by the IASI footprint of tens of kilometres), would cause an underestimation of the model SO2 burden compared to IASI. This effect will become more pertinent with dilution over time as the SO2 column approaches the 0.3 DU limit."

*page 11 line 13/14:*

*Do you mean the maximum on 0.9 Tg here? Please clarify.*

*The text was changed to:*

"For these calculations we choose the $SO_2$ burden maximum as the initial value (0.9 Tg in our study)."

*page 11 line 15-page 12 line 2:*

*Jumping between your results and the findings of Haywood confused me. Consider presenting your results first and compare then with the results of Haywood.*

We propose the following reworked text:

"The e-folding time-constant for $SO_2$ is approximately 17.0 days for the simulation including HCl, about two days longer than the approximately 15.0 days for the simulation that was run without HCl. When these CESM1(WACCM) model outputs are adjusted to correct for the 0.3 DU $SO_2$ lower value of the particular IASI retrievals used they yield e-folding time-constants of 11.5 and 10.0 days, respectively. For the IASI $SO_2$ retrieval of Clarisse et al. (2012) we calculate 12 days, i.e. very similar to the adjusted model simulation with SO2 and HCl co-injection (11.5 days). For comparison, Haywood et al. (2010) report that the HadGEM2 model yields a 13–14-day $SO_2$ e-folding time (assuming a higher $SO_2$ injection of 1.2 Tg and no HCl co-injection). Regarding IASI observations, Haywood et al. (2010) report an IASI $SO_2$ e-folding time of 10–11 days, whilst using our method we calculate 9 days for the IASI retrieval of 2010. This is summarised in Table 2."

*page 12 table 2:*

*What is the significance of your e-folding time? All results are presented as integers, but the one for your model run with HCl and IASI detection threshold says 11.5 days.*

We added a decimal naught in our results in order to clarify the precision of the computed e-folding times.

*page 12 line 16:*

*Can you quantify the "good general agreement"? Is the agreement in the upper panel of Figure 4 within the error of the OPC and the uncertainty of your volcano-off simulation?*

*page 13 figure 4:*

*I suggest to add the measurement errors (that are given in Section 2.2) to the OPC data. Without them it is really difficult to judge if the simulations and observations agree quantitatively within their errors on the logarithmic scale. Further, can you indicate the uncertainty range of the simulations?*

We added the OPC error-bars on Fig. 4. The "uncertainty" of the model runs is something very difficult to estimate: it would depend on the dynamical processes, chemical processes, resolution used, etc. and would be very difficult to estimate on single-configuration runs. We did not carry out ensemble simulations, which were out of the scope of this study. The resolution we used was somewhat classical: it would have been cumbersome to lower it, and furthermore it is easily comparable to those of the studies we used as references.

*page 14 line 2:*

*In Section 3.2 (page 10) you mentioned that your model is too disperse. It seems to me that here it is the most likely source of error. I'd suggest to compare with your IASI observations as in Fig. 1 and add this to Figure A1. You could also compare with Wu et al., ACP, 2017 and discuss.*

IASI only sees $SO_2$, not sulphate particulates, and it is too late in time to compare $SO_2$ over Laramie on the dates considered ($SO_2$ is too dispersed to be observed). As for the comparison with Wu et al., 2017, 22[nd] June is not readily readable on Wu et al.'s plot, and furthermore it is an $SO_2$ map, not sulphates. Instead we do now earlier mention the $SO_2$ results of Wu et al. (2017) in relation to our Figure 1.

*page 14 line 13:*

*How do you know that the CN particle mode has ever been different from the volcano-off simulation over Laramie?*

We here refer to the simulations, and the CN mode has been different throughout time. We reformulated the sentence as:

"... indicating the progressive return of the simulated concentrations to background conditions for this size range".

*page 14 line 25-30:*

*You discuss the discrepancies between the measurements and your model results at altitudes below*

*and above your injection height. I assume that not injecting SO2 below 10 km and above 15 km also contributes to the differences. I suggest to add this to the discussion.*

Concerning the possible injection of $SO_2$ below 10 km of altitude, we disagree with the referee: because of wash-out processes, it is impossible to maintain a tropospheric signal after such a long period of time. As for the injection above 15 km, please refer to our answers to the major comments; a substantial injection at these altitudes remains to be clearly proven (see e.g. Mattis et al. (2010), Doeringer et al. (2012), Jégou et al., (2013)). We add text to to the manuscript discussion-conclusion about uncertainties in injection altitude, see our response below to the comment for *page 25 line 9-11.*

*page 15 figure 5:*

*Why are you using different colors for similar size bins (e.g. top: 885 nm is orange, bottom, 850 nm is red)? Why are there 3 size bins below 440 nm on 18 May 2010 but the other profiles start with 440 nm? I suggest to merge the lowest size bins for 18 May to make it comparable to the measurements in August. Please indicate the measurement and simulation uncertainty. On a logarithmic scale it's really hard to tell if there is a good agreement. Please also optimize the colors. In the 2 bottom panels there are two indistinguishable green lines.*

Size bins for STAC change according to the calibration process, and therefore they can differ for different flights. Our principal point is the consistency between measurements and model computations. We added the error values on the total STAC counts (+/-6%) in the legend; for clarity reasons, we chose not to add them on the plots directly, which would make the figures unreadable.

*page 16 line 1:*

*Has there ever been a comparison between STAC and OPC that might explain the difference?*

Yes, indeed, Renard et al., Applied Optics, 2002, provides such a comparison. This paper shows a good accordance between the instruments.

Jean-Baptiste Renard, Gwenaël Berthet, Claude Robert, Michel Chartier, Michel Pirre, Colette Brogniez, Maurice Herman, Christian Verwaerde, Jean-Yves Balois, Joëlle Ovarlez, Henri Ovarlez, Jacques Crespin, and Terry Deshler, *Optical and physical properties of stratospheric aerosols from balloon measurements in the visible and near-infrared domains. II. Comparison of extinction, reflectance, polarization, and counting measurements*, Appl. Opt. 41, 7540-7549 (2002).

We added a sentence p. 7, l. 18: "Note that both STAC and University of Wyoming OPCs have been compared in Renard et al. (2002)."

*page 16 line 1-3:*

*Does your model simulation suggest coagulation, condensation and sedimentation? What about transport to lower latitudes and dilution between August and November? What is the sedimentation speed and distance of e.g. 0.5 μm particles over 3 months? Shouldn't they show up at a lower*

*altitude in the OPC data then? Please substantiate your explanation.*

The model does indeed accounts for coagulation, condensation and sedimentation. It is very difficult to address the referee's question, since all processes (transport, coagulation, sedimentation) act together within the model. It appears hard to disentangle each of them, unless one carries out separate simulations activating processes one by one.

To answer the example question: the sedimendation speed and distance vary with the altitude of injection. Hamill et al. (1997) state that a 0.5 µm particle injected e.g. at 22 km of altitude has a sedimentation speed of 0,005 cm/s, i.e. 133 m/mth, which is slow.

We added reference to the transport and dilution processes in the revised version of the manuscript:

"This indicates the result of coagulation, condensation, sedimentation, and transport and dilution processes"

*page 16 line 32/33:*

*How do you estimate the local tropopause? Do you use the thermal or dynamical tropopause? Which PVU threshold? What do you use at the pole/equator? What is the uncertainty of your tropopause? Please provide details.*

According to WACCM's documentation, the local tropopause is calculated using the WMO lapse rate definition (thermal definition).

We erroneously stated that dynamical tropopause was considered: this was corrected in the revised version of the manuscript.

Tropopause levels are aligned to the altitude levels of the model, so that the possible error follows steps of ~1 km in altitude.

*page 17 figure 6:*

*As I understood, the main purpose of this figure is to compare the STAC measurements with the model simulations, I suggest to select a smaller range on the y-axis so that it fits to the STAC data (e.g. dN 1e-3-1e2, dV, 1e-15-1e-12, dV 1e-21-1e-17). In the present figure I can only see that the no-volcano runs do not fit. Further, I consider error bars on the STAC measurements helpful.*

We reworked the figure as suggested, with the y-axis ranges given. The error bars were not included on the figures for the sake of clarity, but mention to an error on the totals of +/-6% was added to the legend.

*page 18 line 4-6: Why don't you rely your Zmin(Φ,λ,t) not solely on your analyses of 2009 shown in Fig A2? I consider a 2009 histogram more appropriate than a 2012 histogram with corrections.*

Actually, the histogram considered was indeed over the period 2009—2010. The manuscript was corrected accordingly.

*page 18 line 6.*

*Ok, it's dynamical tropopause. Which PVU is your tropopause? What do you use in the tropics/at the equator? 380 K? Thermal tropopause? What is the accuracy of your tropopause? Please provide details.*

Please refer to our answer to a previous point concerning page 16, lines 32—33. We erroneously assumed a dynamical tropopause was computed. As a matter of fact a thermal definition is used within the model, compliant with the WMO definition.

*page 28 Figure A2:*

*Please extent the y-Axis to accommodate all data points and provide information on the color code. Does the black line mean that you used only a Zmin(Φ) for your model degradation and not a Zmin(Φ,λ,t) as described on page 18? To me it seems that there is some seasonality. Would your analysis improve if you used a Zmin(Φ,t)? At high and very low latitudes (0-10N, 50-90N) the minimum altitude threshold seems to be below the median of the data points. What is your reason not using the median?*

The y-axis was extended and some comment on the colour code was added to the legend.

To provide a slight correction: the black line is not $Z_{min}$, but $\Delta$. For simplicity reasons, we did not consider any time dependency, and we deem the results obtained fair enough using this assumption.

As for the possible use of a mean or a median: our main goal was a demonstration of model degradation as a better approach to compare to observations, without going into an unnecessary detail of pixel-by-pixel correction. Therefore, though the derivation of $\Delta$ could of course have been carried out linearly regressing the OSIRIS data, we chose a qualitative approach, showing the first order effect. Furthermore, we were notified by personal communication (Adam Bourassa, Univ. Saskatchewan), that the OSIRIS sets of data were most likely to be updated in a near future, making the relevance of older versions of the data quite relative.

*page 18 figure 7:*

*I suggest checking seasonality for your degradation. Unfortunately I cannot tell from Fig. A2 in which months your degradation altitude fits best, but at high latitudes you have a good agreement in October, November, April, and May and at low latitudes (0-20N) you have the yellow (day 250-350) and blue (day 425-525) features that might coincide with your data points above and below your Zmin(Φ). Please clarify.*

As already stated, our degradation algorithm does not consider any time dependency. Due to the extended use of the Mie scattering code, it would have been cumbersome to check the effect of seasonality. This effect is possible, and likely, but was not investigated in the current study.

*page 20 line 1-3:*

*I don't understand what you mean. Please detail where and to what extent the anomalies in Fig 8 agree better than the SAODs in Fig 7. Except from the shaded area indicating OSIRIS measurement gaps in the polar region in the middle panel of Fig. 8 I cannot see obvious additional information. Figure 7 already shows impressively that OSIRIS misses a substantial fraction of lower stratosphere sulfate aerosol.*

The aim of calculating the anomalies is to set aside the background conditions, and to focus on the volcano effects only. We deem important to show that both the total SAODs and the anomaly optical depths match: the former is for comparison with previous studies, and the latter for the quantification of the volcanic effect.

*page 20 figure 9:*

*Why are you showing 550 nm extinction here? It is not used anywhere else, all other OSIRIS data is presented for 750 nm. Please clarify and consider using less colors (7 might be sufficient) in the bottom figure. Some are indistinguishable.*

550 nm is the standard output wavelength in CARMA; it is used here just for validation of the time evolution, and we chose to consider this readily exploitable output. Like for Fig. 1, we choose not to alter the number of colours, in order to keep a good rendition of the dynamics range.

*page 21 line 3-5:*

*Please specify what you mean with "strongest measurement-biases shortly after the eruption". Do you mean OSIRIS high Zmin, or its saturation, or its rather coarse sampling that might miss local maxima of the plume filaments shortly after the eruption? Perhaps you want to compare with Günther et al. , ACPD (2017) Fig. 6, which is similar to your Fig. 9, but with different model and satellite data.*

By "strongest measurement biases", we refer to the saturation process. This was clarified in the text. As for the comparison with Günther et al. (2017): our Fig. 9 is modelled extinction, whereas Günther et al.'s Fig 6 is modeled sulfur mass. Furthermore, the $SO_2$ mass is underestimated by MIPAS in first month.

*page 21 table3, line 12-19:*

*For which purpose do you present e-folding times from other studies? They are not discussed here.*

We present these values for quantification purposes. The noticeable result is the agreement between our degraded model and OSIRIS measurements.

The goal is not to discuss these in detail (they can depend firstly on many factors in the model, and also on how e-folding times are calculated, what period, etc.), but we do highlight that "One can note that the e-folding times vary quite significantly between authors, including those computed

from OSIRIS's data (it is likely that different versions of the OSIRIS data —v.5.05 up to v.5.07 for the present study—were used)".

*page 22 line 18/19:*

*Please add a reference for those removal processes.*

We added the following reference:

Hamill, P., Jensen, E. J., Russell, P. B., & Bauman, J. J. (1997). The life cycle of stratospheric aerosol particles. *Bulletin of the American Meteorological Society*, *78*(7), 1395-1410.

*page 22 line 25-28:*

*You might want to include the reff retrieval by Doeringer et al. (2012), who found 0.1—0.3 µm for the Sarychev, into your discussion.*

This was added to the text.

"The latitudinal trend in reff simulated by our model is broadly consistent with the trend reported from ground-based remote sensing at Eureka (Nunavut, Canada) that found reff = 0.29 µm (O'Neill et al. (2012)), and with ACE measurements, which report reff = 0.1—0.3 µm (Doeringer et al. (2012)).

*page 23 figure 10:*

*Please reduce number of colors. There are too many indistinguishable shades of red and green.*

We choose to keep the number of colours as is, notably to maintain a good dynamic range.

*page 22 line 34:*

*Which chlorine and bromine species do you mean?*

They are $ClO_x$ and $BrO_x$. This was added to the text in the revised version of the manuscript.

*page 23 line 2:*

*Please note, the washout is not necessarily as efficient as in the Pinatubo case (von Glasow, Chemical Geology, 2009).*

We agree with the referee. We add the clause "... though the washout for the Sarychev case was not necessarily as efficient [von Glasow, 2009]."

*page 23 line 20 - page 24 line 2:*

*I did not understand this sentence. Please fix it.*

We corrected the sentence to:

"This is primarily due to the higher aerosol loadings in these regions, and to favorable solar illumination conditions for which the catalytic ozone loss cycles (through OH radical production) are enhanced (Berthet et al., 2017)."

*page 24 line 2-5:*

*I did not understand this sentence. Please reword and provide a reference.*

We lightened the text, and replaced the concerned sentences with:

"A more detailed description of involved chemistry processes is provided in [Berthet et al., 2017]."

*page 24 line 5:*

*I suggest starting a new paragraph here to clearly differentiate between heterogeneous reactions on aerosol particles and PSC particles. Isn't HCl the main reservoir of Cl and not ClONO2? There is HNO3 uptake by PSC particles that sediment out and hence lead to denitrification. I suggest to explicitly mention PSCs in this process and to reword this sentence.*

As previously stated, the concerned paragraph was replaced.

It is not the scope of this paper to describe stratospheric chemistry linked to PSC.

*page 24 line 16:*

*is 5% versus 7% a and 50 versus 60% a significant difference in your model? What is the uncertainty?*

We cannot properly speak of uncertainty, since the model runs with a deterministic calculation (a large model ensemble study is beyond the scope of our work).

*page 25 line 8:*

*I did not find your results convincing that an injection altitude of 11 to 15 km is realistic. I'd rather interpret your results that there are discrepancies between simulations and measurements above 15 km (see Fig. 4). Further your lower SAOD in the degraded model data compared to OSIRIS (Fig. 7 and 8) may be a result of not accounting for the SO2 injections into altitudes above 15 km that have been observed by several independent measurements.*

We removed the clause "appears to be realistic", which we agree was too strong an assertion.

We added the following sentence: "The lack of more resolved data might be a source of uncertainty on the injection altitudes."

*page 25 line 9-11:*

*Comparing your plume simulation in Fig 1 to IASI data and to the model simulations and AIRS data in Wu et al., ACP (2017), I find some shortcomings in this approach, which become also visible in several details and interpretations on which I commented before. Hence, I suggest to rephrase this sentence and add some discussion on potential errors due to the injection assumption.*

We propose the following text change:

"The lack of more resolved data might be a source of uncertainty on the injection altitudes. However, this overall quantitative agreement reflects the model performance in $SO_2$ oxidation, atmospheric dispersion and aerosol processing. It indicates a suitable choice of eruption source parameters as used in previous studies e.g. Haywood et al. (2010) (an injection altitude ranging from 11 km to 15 km for $SO_2$, a vertical even spread of the total mass of gases injected, and a sole injection of the total gas mass on 15 June 2009, neglecting other minor injections on other days). These eruption source parameters did provide good results. They might need to be refined for model studies at higher temporal or spatial resolution, see Wu et al. (2017), Gűnther et al. (2017). We point out that an injected mass of 0.9 Tg SO2 (Clarisse et al., 2012; Realmuto and Berk, 2016) instead of 1.2 Tg of previous studies e.g. Haywood et al. (2010) is a fair hypothesis, and enables the model to closely reproduce the observed $SO_2$ burden according to the IASI retrievals of Clarisse et al. (2012)."

*page 26 line 29-31: In which respect is this statement different from the findings in Ridley et al. (2014)?*

We propose to keep this sentence as it is, because the findings of Ridley et al. (2014) (and similarly Mills et al., 2016) are more general in terms of temporal scale and volcanoes, are not directly related to OSIRIS SAOD, and only consider minimum altitude not the saturation effects in observation biases. Our study is focused on Sarychev eruption specifically, and presents a degradation approach to the model output (adjusted) to better compare to OSIRIS measurements.

We keep the sentence: "Our study therefore highlights that previous modelling studies (involving assumptions on particle size) that reported agreement to (biased) post-eruption estimates of SAOD derived from OSIRIS likely underestimated the climate impact of the 2009 Sarychev Peak eruption." but we propose to insert in the Introduction, page 3, line 32:

"More generally, underestimation of SAOD due to neglect of lower stratospheric volcanic aerosols has also been highlighted by Kravitz et al. (2011), Ridley et al. (2014), Andersson et al. (2015), Mills et al. (2016). As model studies of Sarychev eruption to date…"

**Technical Suggestions**

- *page 1, line 19: confirm*

This was corrected.

- *page 5 line 5: please sort references chronologically*

This was corrected.

- *page 5 line 7, 8: references for Sindelarova and Kettle are missing*

The concerned references were added to the reference list.

- *page 6 line 22: just write IASI*

This was corrected.

- *page 6 line 31: Define abbreviation at first usage only. Please also check in other places e.g. for OSIRIS page 7 line 20.*

These were corrected.

- *page 7 line 5: What does StraPolEte stand for? Is there any reference to the AEROWAVE" project?*

The acronym for StraPolÉte was already defined in the text.

We unfortunately have no paper reference for the AEROWAVE project.

- *page 10 figure 2: In the figure the IASI retrieval is green, but the caption says light blue.*

This was corrected.

- *page 11 line 10: "... Section 3.6." Please start a new paragraph here.*

A new paragraph was started there.

- *page 13 figure 4 caption: "solid" instead of "full" line*

This was corrected.

- *page 14 line 7: "... observations." Please start a new paragraph here.*

A new paragraph was started there.

- *page 14 line 12: Do the profiles only appear to be close to each other or are they close?*

They are close. This was corrected in the text.

- *page 18 line 15: Do you mean: A comparison for these months is therefore impossible?*

We replaced "difficult" by "not possible".

- *page 21 line 7: (9) = (Fig. 9)?*

This was corrected.

- *page 21 line 21/22: Please replace "... elsewhere, notably ..." by e.g. and refer to Table 3.*

This was corrected, and a reference to Table 3 was added.

- *page 22 line 17: ... , which*

This was corrected.

- *page 22 line 22: material*

This was corrected.

- *page 22 line 29: ...are thus a major ...*

This was corrected.

- *page 22 line 33: "... investigating the impacts of modern day eruptions on stratospheric..." Please fix this sentence.*

We rewrote the sentence as: "Most studies investigating the impacts of modern day eruptions on stratospheric chemistry..."

- *page 24 line 30: Please write "... over one day of eruption ..."*

This was corrected.

- *page 25 line 5: Please write "... suggest that the effective radius becomes ..."*

This was corrected.